

**CO₂ effects on diatoms: A Synthesis of more than a decade of ocean**
**acidification experiments with natural communities**
**Lennart. T. Bach[1,2*], Jan. Taucher[1]**
[1]Biological Oceanography, GEOMAR, Helmholtz Centre for Ocean Research, Kiel,
Germany
[2]Institute for Marine and Antarctic Studies, University of Tasmania, Hobart, Tasmania,
Australia
*Corresponding Author lennart.bach@utas.edu.au
Keywords: Ocean acidification, climate change, mesocosm, literature review, food web,
phytoplankton, Bacillariophyceae
**Abstract**
Diatoms account for ~40% of marine primary production and are considered to be key
players in the biological carbon pump. Ocean acidification (OA) is expected to affect
diatoms primarily by changing the availability of $CO_2$ as a substrate for photosynthesis
or through altered ecological interactions within the marine food web. Yet, there is little
consensus how entire diatom communities will respond to increasing $CO_2$. To address
this question, we synthesized the literature from over a decade of OA-experiments with
natural diatom communities to uncover: 1) if and how bulk diatom communities respond
to elevated $CO_2$; 2) if shifts within the diatom communities could be expected and how
they are expressed with respect to taxonomic affiliation and size structure. We found that
diatom communities responded to high $CO_2$ in ~60 % of the experiments and in this case
more often positively (56 %) than negatively (32 %; 12 % did not report the direction of



change). Shifts among different diatom species were observed in 65 % of the experiments.
Our synthesis supports the hypothesis that high $CO_2$ particularly favors larger species as
12 out of 13 experiments which investigated cell size found a shift towards larger species.
Unraveling winners and losers with respect to taxonomic affiliation was difficult due to
a limited database, but there is evidence that the genus *Pseudo-nitzschia* could be among
the losers. We conclude that OA-induced changes in diatom competitiveness and
assemblage structure must be classified as a "risk for ecosystem services" due to the
pivotal role diatoms play in trophic transfer and biogeochemical cycles.
**1. Introduction**
The global net primary production (NPP) of all terrestrial and marine autotrophs amounts
to approximately 105 petagrams (Pg) of carbon per year (Field et al., 1998). Marine
diatoms, a taxonomically diverse group of cosmopolitan phytoplankton, were estimated
to contribute up to 25 %  (26 Pg C year$^{-1}$) to this number, which is more than the annual
primary production in any biome on land (Field et al., 1998; Nelson et al., 1995; Tréguer
and De La Rocha, 2013). Thus, diatoms are likely the most important single taxonomic
group of primary producers on Earth and any change in their prevalence relative to other
phytoplankton taxa could profoundly alter marine food web structures and thereby affect
ecosystem services such as fisheries or the sequestration of $CO_2$ in the deep ocean
(Armbrust, 2009; Tréguer et al., 2018).
The most conspicuous feature of diatoms is the formation of a silica shell, which is
believed to primarily serve as protection against grazers (Hamm and Smetacek, 2007;
Pančić and Kiørboe, 2018). Since the formation of this shell requires dissolved silicate,
diatoms are often limited by silicon as a nutrient rather than by nitrogen or phosphate
(Brzezinski and Nelson, 1996). However, when dissolved silicate is available, diatoms



benefit from their high nutrient uptake and growth rates, allowing them to outcompete
other phytoplankton and form intense blooms in many ocean regions (Sarthou et al.,

51    2005).

Diatoms display an enormous species richness, with recent estimates being in the range
of 30,000 species (Mann and Vanormelingen, 2013). Although only a fraction has been
morphologically described, known diatom taxa span a size range of several orders of
magnitude (<5 μm up to a few mm) with a wide range of morphologies and life strategies,
e.g. single cells and cell chains, pelagic and benthic habitats (Armbrust, 2009; Mann and
Vanormelingen, 2013; Sournia et al., 1991). Accordingly, they should not be treated as
one functional group, but rather as a variety of subgroups occupying different niches.
It is well recognized that the global importance of diatoms as well as their diversity in
morphology and life style is tightly linked to the functioning of pelagic food webs and
elemental cycling in the oceans. For example, iron enrichment experiments in the
Southern Ocean found that a shift in diatom community composition from thick- to thin-
shelled species ("persistence strategy" vs. "boom-and-bust strategy") can enhance carbon
and alter nutrient export via sinking particles (Assmy et al., 2013; Smetacek et al., 2012).
This may not only affect element fluxes locally but enhance nutrient retention within the
Southern Ocean and reduce productivity in the north which underlines how important
diatom community shifts can be on a global scale (Boyd, 2013; Primeau et al., 2013;
Sarmiento et al., 2004). Likewise, the cell size of diatoms can play an important role in
transferring energy to higher trophic levels, as the dominance of larger species is
generally considered to reduce the length of the food chain and lead to higher trophic
transfer efficiency (Sommer et al., 2002). Consequently, understanding impacts of global
change on diatom community composition is crucial for assessing the sensitivity of
biogeochemical cycles and ecosystem services in the world oceans.




It has become evident that the sensitivity of diatoms to increasing pCO$_2$ is highly variable,
likely being related to specific traits such as cell size or the carbon fixation pathway, as
well as interactions with other environmental factors such as nutrient stress, temperature
or light (Gao et al., 2012; Hoppe et al., 2013; Wu et al., 2014). However, it is still rather
unclear how these species-specific differences in CO$_2$ sensitivities manifest themselves
on the level of diatom communities. This knowledge gap motivated us to compile the
presently available experimental data in order to reveal common responses of diatom
communities to high CO$_2$ and thereby assess potential scenarios of shifts in diatom
community composition under ocean acidification.
**2. Literature investigation**
**2.1. Approach**
Our original intention was to conduct a classical meta-analysis, which would have yielded
the benefit of a quantitative measure of diatom responses to OA, expressed as an overall
effect size (i.e. combined magnitude) such as the response ratio. However, our literature
analysis revealed a large variability in experimental pCO$_2$ ranges as well as measured
response variables, which cannot be directly compared among each other (e.g.
microscopic cell counts, pigment concentrations, genetic tools). These limitations impede
data aggregation as required for a classical meta-analysis. Furthermore, experimental
setups differed widely in terms of other environmental factors such as temperature, light,
and nutrient concentrations, all of which are known to modulate potential responses to
pCO$_2$ (Boyd et al., 2018), thereby further complicating data aggregation for meta-
analysis. Therefore, we chose an alternative, semi-quantitative approach where diatom
responses to increasing CO$_2$ are grouped in categories (see section 2.2) and also allows
to account for differences in experimental setups, e.g. with respect to container volume.



While this approach excludes the determination of effect size, it provides an unbiased
insight on the direction of change of potential $CO_2$ effects.
**2.2. Data compilation**
We explored the response of diatom assemblages to high $CO_2$ (low pH) by searching the
literature for relevant results with Google Scholar (December 15, 2017) using the
following search query: diatom OR Bacillariophyceae AND "ocean acidification" OR
"high $CO_2$" or "carbon dioxide" OR "elevated $CO_2$" OR "elevated carbon dioxide" OR
"low pH" OR "decreased pH". The first 200 results were inspected and considered to be
relevant when they were published in peer-reviewed journals, contained a description of
the relevant methodological details, a statistical analysis or at least a transparent
description of variance and uncertainties, and tested $CO_2$ effects on natural plankton
assemblages (artificially composed communities were not considered). We then carefully
checked the cited literature in these relevant studies to uncover other studies that were
missed by the initial search. Furthermore, we checked the "Ocean Acidification news
stream provided by the Ocean Acidification International Coordination Centre" under the
tag "phytoplankton" (https://news-oceanacidification-icc.org/tag/phytoplankton/) for
relevant updates since December 2017 (last check on January 16, 2019).
There were two response variables of interest for the literature compilation:
1) The response of the "bulk diatom community" to high $CO_2$. For this we checked if the
abundance of diatoms, the biomass of diatoms, or the relative portion of diatoms within
the overall phytoplankton assemblage increased or decreased under high $CO_2$ relative to
the control. We distinguished between "positive", "negative", and "no effect" following
the statistical results provided in the individual references. When the $CO_2$ effect on the
bulk community was derived from abundance data we also checked if there are



indications for a concomitant shift in the biomass distribution among species. This is
relevant because, for example, an increase in bulk abundance could coincide with a
decrease in bulk biomass when the species driving the abundances is smaller. We found
no indications for conflicting cases but acknowledge that not every reference provided
sufficient data on morphological details to fully exclude this scenario.
2) The $CO_2$-dependent species shifts within the diatom community with respect to
taxonomic composition and/or size structure. Unfortunately, cell size of the species was
not reported for all experiments. Thus, we distinguished between "no shifts", "shifts
between species with unspecified size", as well as "shifts towards larger or smaller
species" when this information was provided. Furthermore, we noted the winners and
losers within the diatom communities when these were reported (on the genus level).
In case the data was taken from factorial multiple stressor experiments (e.g. $CO_2$ x
temperature) we only considered the control treatment at ambient conditions (e.g. at
control temperature). Furthermore, we extracted various metadata from each study
largely following the literature analysis of (Schulz et al., 2017). All bulk diatom
responses, community shifts, and metadata is compiled/described in Table 1 and most of
it is self-explanatory (e.g. incubation temperature). The habitats of the investigated
diatom communities were categorized according to water depth, salinity, or life style in
the case of benthic communities: "oceanic" = water depth > 200 m (unless the habitat lies
within a fjord or fjord-like strait), S > 30; "coastal" = water depth < 200 m, S > 30;
"estuarine" = water depth < 200 m, S < 30; "benthic" = benthic communities (diatoms
growing on plates) were investigated. We reconstructed the water depth in case it was not
provided in the paper using Google Earth Pro (version 7.3.2.5495). The coordinates
provided in some of the experiments conducted in land-based facilities were imprecise
and marked positions on land. In this case the habitats were set to coastal or estuarine



depending on salinity. If salinity was not given we checked the location on Google Earth
for potential fresh water sources and also checked the text for more cryptic indications
(e.g. "euryhaline" in a lagoon were strong indications for an estuarine habitat). The
methods with which responses of the bulk diatom communities to high OA were
determined varied greatly among studies and included light microscopy (LM), pigment
analyses (PA), flow cytometry (FC), genetic tools (PCR), and biogenic silica (BSi)
analyses (Table 1).
**2.3. Balancing the influence of smaller and larger scale experiments to account for**
**the "degree of realism"**
The most realistic OA experiment would be one where all aspects of the natural habitat
are represented correctly. Such setups are possible for benthic communities which can be
sampled *in situ* along a natural $CO_2$ gradient at volcanic $CO_2$ seeps (Fabricius et al., 2011;
Hall-Spencer et al., 2008; Johnson et al., 2011). However, this does not work easily for
the large majority of pelagic studies compiled herein due to water advection. Thus, OA
experiments with pelagic communities are performed in closed containers which
inevitably cause artefacts (Calvo-Díaz et al., 2011; Ferguson et al., 1984; Guangao, 1990;
Menzel and Case, 1977). However, the degree by which they are unrealistic will differ
from study to study depending on the experimental design (Duarte et al., 1997). Here, we
aimed to develop a metric that allows us to estimate the realism of experiments with
pelagic communities in order to balance their influence on the final outcome of the
literature analysis. Most certainly, we do not mean to devalue any studies but think that
the highly different scales of experiments should not be ignored when evaluating the
literature. In the following we will first derive the equation for the proposed metric –
termed the "relative degree of realism (RDR)" – and introduce the underlying
assumptions. Afterwards we describe aspects that were considered while conceptualizing



the RDR.
The experimental design in the studies considered herein ranged from smaller bottle
experiments (e.g. 1 L) to *in situ* mesocosm studies with considerably larger incubation
volumes (e.g. 75000 L). While smaller differences in incubation volumes (e.g. 0.5 vs. 2
L) were shown to have no, or a minor, influence on physiological rates (Fogg and
Calvario-Martinez, 1989; Hammes et al., 2010; Nogueira et al., 2014; Robinson and
Williams, 2005), they can influence food web composition e.g. by excluding larger
grazers (Calvo-Díaz et al., 2011; Spencer and Warren, 1996). Larger differences of
incubation volumes (e.g. 10 vs. 10000 L) are considered to be important in all aspects
(Duarte et al., 1997), with the larger volume being more representative of natural
processes (Sarnelle, 1997). Therefore, our first assumption to conceptualize the RDR was
that larger incubation volumes represent nature generally better than smaller ones.
Plankton communities were pre-filtered in many experiments to exclude larger and often
patchily distributed organisms. This is a valid procedure to reduce noise and to increase
the likelihood to detect $CO_2$ effects but it also influences the development of plankton
communities as they modify the grazer/prey link within the food web (Ferguson et al.,
1984; Nogueira et al., 2014; Pomeroy et al., 1994). For example, (Nogueira et al., 2014)
compared plankton successions of pre-filtered (100 µm) and unfiltered communities and
found that the removal of larger grazers and diatoms gave room for green algae and
picophytoplankton to grow. Such manipulations make the experiment less representative
for a natural food web which brought us to the second assumption for the RDR: The
smaller the mesh size during the pre-filtration treatment, the less complete and thus the
less realistic is the pelagic food web.
To parameterize the two abovementioned assumptions we first converted the volume



information provided in each experiment into a volume-to-surface ratio (V/S). The
underlying thought is that V increases with the third power to the surface area of the
incubator and is indicative for the relation of open space to hard surfaces (Ferguson et al.,
1984). Therefore we first converted V into a radius (r) assuming spherical shape:
$r = \sqrt[3]{\dfrac{3}{4}\dfrac{V}{\pi}}$              (1).
The surface (S) of the spherical volume was calculated as:
$S = 4\pi r^2$              (2)
Assuming spherical shape was necessary because there is generally no information about
the shape of the incubation containers available. Although shape can influence processes
within the container (Pan et al., 2015), it is probably a less important factor to consider in
light of the large volume differences compared herein (Table 1).
The influence of pre-filtration treatments of the investigated plankton community is
implemented by multiplying the V/S with the third root of the applied mesh size ($d_{mesh}$ in
µm) so that the RDR is defined as:
$RDR = \dfrac{V}{S}\sqrt[3]{d_{mesh}}$              (3).
Thus, as for V/S, the influence of $d_{mesh}$ on RDR does not linearly increase but dampens
with increasing $d_{mesh}$. The rationale for the non-linear increase is that incubations will still
have an increasing bias even if they do not have any pre-filtration treatment due to
generally increasing organism motility with size. For example, when collecting a
plankton community with a Niskin bottle, more motile organisms can escape from the
approaching sampler so that the food web composure is still affected even without



subsequent pre-filtration. For this reason we also capped the maximum $d_{mesh}$ to 10,000
µm when there was no pre-filtration treatment applied since none of the studies included
significantly larger organisms. Figure 1 illustrates the change of RDR as a function of V
and $d_{mesh}$. High RDRs are calculated for large-scale *in situ* mesocosm studies (~50 – 190)
while bottle experiments yield RDRs between ~1 – 12.
The key pre-requisite for an experimental parameter to be included in the RDR equation
(eq. 3) was that it is reported in all studies. Many parameters that we would have liked to
use for the RDR are either insufficiently reported (e.g. the light environment) or not
provided quantitatively at all (e.g. turbulence). We therefore had to work with very basic
properties related to the experimental setup rather than to the experimental conditions.
A particularly critical aspect of the RDR we had to deal with was the duration of the
experiments (Time). Time is a quantity, which is reliably reported in all studies and
therefore principally suitable for the RDR. Our first thoughts were that a realistic
community experiment should be long enough to cover relevant ecological processes
such as competitive exclusion and therefore also parameterized Time in the first versions
of the RDR equation. However, we decided to not account for it in the final version
because the factors that define the optimal duration of an experiment are poorly
constrained. For example, a 1 day experiment in a 10 L container could indeed miss
important $CO_2$ effects caused by food web interactions. On the other hand, a 30 days
experiment in the same container could reveal such indirect effects but at the same time
be associated with profound bottle effects and make the study unrepresentative for
simulated natural habitat. Thus, too long and too short are both problematic and the
optimum is hard to find. One such attempt to find the optimum was made by (Duarte et
al., 1997) who analyzed the plankton ecology literature between 1990 – 1995. By
correlating the experimental duration with the incubation volume of published





experiments they provided an optimal length for any given volume. However, as noted
by (Duarte et al., 1997), their correlation is based on publication success and therefore
rather reflects common practice in plankton ecology experiments and not necessarily a
mechanistic understanding of bottle effects. Thus, as there is no solid ground for a
parameterization of Time we ultimately decided to not consider it for the RDR.
**3. Results**
We found 54 relevant publications on $CO_2$ experiments with natural diatom assemblages.
Some publications included more than one experiment so that 69 experiments are
considered hereafter (Table 1). Most were done with diatom communities from coastal
environments (46 %) and oceanic (28%) environments. Estuarine and benthic
communities were investigated in 16 % and 6% of the studies, respectively (Figs. 2 and
3). 4 % of the studies did not provide coordinates where the samples were taken although
the region was reported (Table 1; Fig. 3).
One third (33 %) of all experiments revealed a positive influence of $CO_2$ on the "bulk
diatom community" (see section 2.2),  while 19 % revealed a negative one. 7 % of the
studies found a $CO_2$ effect but did not specify whether it is a positive or negative one. 41
% found no effect (Fig. 4A; left column). Those experiments that revealed positive $CO_2$
effects on bulk diatom communities yielded the highest cumulative RDR score ($\sum$RDR)
of 605 while the $\sum$RDR for negative $CO_2$ effects was 266. No $CO_2$ effects yielded a score
of 768 while an "unspecified effect" yielded 266.
$CO_2$-dependent shifts in diatom species composition were investigated with light
microscopy except for (Endo et al., 2015) who used molecular tools,. Species shifts were
investigated in a subset of 40 of the 69 experiments (Fig. 4B). Within this subset of 40
studies, 12 (30 %, $\sum$RDR = 265) found a shift towards larger diatom species under high





$CO_2$, 1 (2.5 %, $\sum$RDR = 10) found a shift towards smaller diatom species, and 13 (32.5
%, $\sum$RDR = 103) found no $CO_2$ effect on diatom community composition. 14 studies (35
%, $\sum$RDR = 141) reported a $CO_2$-dependent shift but did not further specify any changes
in the size-class distribution (Fig. 4C).
A taxon-specific assessment of potential winners and losers (on the genus level) was
possible only to a limited extent, because most genera were not present in enough
experiments to get useful results. Only *Chaetoceros*, *Cylindrotheca,* and *Pseudo-*
*Nitzschia* were explicitly investigated in at least 5 experiments, which we set as a
minimum threshold.  *Chaetoceros* responded positively to high $CO_2$ in 6 out of 9
experiments ($\sum$RDR of winning = 84; $\sum$RDR of losing = 61; Fig 5A). *Cylindrotheca*
responded positively in 2 out 5 experiments ($\sum$RDR of winning = 5; $\sum$RDR of losing =
9;Fig. 5B). *Pseudo-Nitzschia* responded positively in 2 out of 9 experiments ($\sum$RDR of
winning = 3; $\sum$RDR of losing = 77; Fig. 5C). Thus, *Pseudo-Nitzschia* is the only genus,
for which there seems to be a fairly consistent negative response to high $CO_2$.
**4. Discussion**
Numerous physiological studies have shown that diatom growth and metabolic rates can
be affected by seawater $CO_2$ concentrations, and that these responses vary widely among
different species (Gao and Campbell, 2014). Such inter-specific differences in $pCO_2$
sensitivity are an important feature as this could alter the composition of diatom
assemblages in a changing ocean. In this regard, it is interesting to note that
paleolimnologists have long been using diatom species composition as paleo-proxy to
reconstruct lake pH (Battarbee et al., 2010). Hence, there is ample evidence that high $CO_2$
conditions have the potential to change the diatom species composition.



Indeed, our analysis revealed that $CO_2$-induced changes in diatom community
composition occurred in 27 out of 40 (i.e. 68 %) of community-level experiments which
investigated species composition (Fig. 4C). This is certainly a conservative outcome
because many studies have only looked at dominant species. In fact, one of the few
experiments that investigated the diatom assemblage with higher taxonomical resolution
found $CO_2$ effects also on subdominant species (Sommer et al., 2015) which may have
been overlooked in many other experiments.
**4.1 Winners and losers in the diatom community**
There was sufficient data (i.e. ≥5 experiments) for the genera *Chaetoceros*,
*Cylindrotheca*, and *Pseudo-Nitzschia* to determine common responses to high $CO_2$.
Among these 3, only *Pseudo-Nitzschia* was fairly consistently identified as a "loser"
within the investigated natural diatom communities. *Chaetoceros* was mostly winning
while *Cylindrotheca* was mostly losing but the trends were not strong. The relatively
weak performance of *Pseudo-Nitzschia* spp. was somewhat surprising because previous
monoclonal experiments with this genus often reported a sometimes pronounced positive
(Sun et al., 2011; Tatters et al., 2012), or no influence of high $CO_2$ on their growth rate
(Sugie and Yoshimura, 2013; Trimborn et al., 2013) but more rarely a negative one
(Tatters et al., 2013). Likewise, laboratory competition experiments between *Chaetoceros*
*debilis* and *Pseudo-Nitzschia subcurvata* saw the latter rather on the winning side under
high $CO_2$ although the difference between them was small (Trimborn et al., 2013). The
reasons for the inconsistency between our results and the impression derived from
controlled laboratory experiments could be manifold. Since our outcome is based on
"only" 9 experiments, it could still be coincidence. However, the pronounced difference
in the RDR value alleviate this concern to some extent (see numbers on top of Fig. 5C).
If the inconsistency is reflecting a true biological pattern than this would emphasize once



more that ecological success within a natural community cannot be easily derived from
physiological studies.

**4.2 CO$_2$ effects on diatom assemblages originating from (direct) physiological**

**responses to high CO$_2$**
Most studies that found effects of pCO$_2$ on diatom communities related these changes to
CO$_2$ fertilization of photosynthesis. Concentrations of CO$_2$ in the surface ocean are
relatively low compared to other forms of inorganic carbon, especially bicarbonate ion
(HCO$_3^-$) (Zeebe and Wolf-Gladrow, 2001). However, RubisCO, the primary
carboxylating enzyme used in photosynthesis, is restricted to CO$_2$ for carbon fixation and
has a relatively low affinity for CO$_2$ compared to O$_2$ (Falkowski and Raven, 2007).
Therefore, diatoms (like many other phytoplankton species) operate a carbon
concentrating mechanism (CCM) to enhance their CO$_2$ concentration at the site of
fixation relative to external concentrations (e.g. by converting HCO$_3^-$ to CO$_2$) and thereby
establish higher rates of carbon fixation than what would be possible when only
depending on diffusive CO$_2$ uptake (Giordano et al., 2005). It is well known that the
proportion of CO$_2$ uptake vs. HCO$_3^-$ uptake for photosynthesis varies largely among
diatoms (Burkhardt et al., 2001; Rost et al., 2003; Trimborn et al., 2008) and is
theoretically also a function of cell size (Flynn et al., 2012; Wolf-Gladrow and Riebesell,
1997). Accordingly, increasing seawater pCO$_2$ may increase the proportion of diffusive
carbon uptake and/or lower the energy and resource requirements for CCM operation
(Raven et al., 2011). From a physiological point of view, these mechanisms could allow
for increased rates of photosynthesis and cell division.
So how do these theoretical considerations align with (A) the variable and species-
specific physiological responses of diatoms to increasing CO$_2$ (Dutkiewicz et al., 2015),



and (B) the results from community-level experiments compiled in this study? Regarding
the variability of physiological responses, progress has recently been made by (Wu et al.,
2014) who experimentally demonstrated a positive relationship between cell volume and
the magnitude of the $CO_2$ fertilization effect on diatom growth rates. Their findings agree
well with theoretical considerations, which predict that high $CO_2$ is particularly beneficial
for carbon acquisition by larger species as they are more restricted by diffusion gradients
due to lower surface-to-volume ratios than smaller cells (Flynn et al., 2012; Wolf-
Gladrow and Riebesell, 1997). The outcome of our literature analysis supports this
allometric concept (Fig. 4, Table 2). Twelve out of 13 experiments in which cell size was
taken into account found a shift towards larger species. This is reflected in the ∑RDR
score of 265 which is ~25 times higher than the opposite result (i.e. $CO_2$-induced shifts
towards smaller diatoms, Fig. 4C). An allometric scaling of $CO_2$ sensitivity is particularly
useful for modelling since cell size is a universal trait which is relatively easy to measure
and therefore frequently available (Ward et al., 2012). Accordingly, it may lead to
significant improvements of ecological and/or biogeochemical model projections under
$CO_2$ forcing when more than one size class for diatoms is considered.
However, although the (Wu et al., 2014) allometric approach constitutes a solid starting
point to help understanding the variable responses of different diatom species, it probably
also still needs some further refinements. For example, central components of CCMs
seem to be adapted to diatom cell sizes, thereby potentially alleviating a strict cell size
dependency of $CO_2$ limitation (Shen and Hopkinson, 2015). Furthermore, size
dependency alone cannot account for taxon-specific differences in the mode of carbon
acquisition (diffusive uptake of $CO_2$ vs. CCM-supported uptake of $HCO_3^-$) and how this
will affect the competitive ability of species under increasing $CO_2$. OA will lead to much
larger changes in dissolved $CO_2$ than in $HCO_3^-$. Thus, species that rely to a larger extent



on a resource-intensive CCM may benefit more from increasing $pCO_2$ on a cellular level,
as they could increase the proportion of diffusive $CO_2$ uptake. However, it is also possible
that the same species would be disadvantaged on the community-level, because their
niche (that is, being competitive at lower $CO_2$ due to an efficient CCM) is diminished
under high $CO_2$ conditions. Which of the scenarios occurs in nature would also depend
on how flexible species are in terms of switching carbon acquisition modes, as well as
resource allocation. In this regard, it is noteworthy that only few physiological studies on
OA effects have taken into account the role of changing nutrient concentrations or even
a transition to nutrient limitation. The available experimental evidence suggests that
increasing $pCO_2$ may reduce cellular nutrient requirements for CCM operations and
therefore free resources for elevated maximum diatom population densities, particularly
when running into nutrient limitation (Taucher et al., 2015). Unfortunately, however, the
relevance of this mechanism has so far only been investigated in monoclonal laboratory
experiments but not on the community-level.
These considerations illustrate that cell size is an important factor, but is not sufficient to
predict physiological or even community-level of diatoms to OA. Moreover, the
allometric concept as well as the additional mechanisms described above generally
presume positive effects of $CO_2$-fertilization, thus yielding no first order explanations for
observed negative responses of diatoms to changing carbonate chemistry. Obviously,
increasing $CO_2$ concentrations are accompanied by increasing proton ($H^+$) concentrations
under ocean acidification. High $H^+$ concentrations may reduce key metabolic rates above
certain thresholds and outweigh the positive influence of $CO_2$ fertilization as has been
observed in coccolithophores (Bach et al., 2011, 2015).
Another pathway by which ocean acidification may alter diatom communities is the pH
effect on silicification and silica dissolution. Low seawater pH should theoretically



facilitate silicification as the precipitation of opal occurs in a cellular compartment with
low pH conditions (pH ~5) (Martin-Jézéquel et al., 2000; Vrieling et al., 1999). At the
same time, a lower pH should reduce chemical dissolution rates of the $SiO_2$ frustule
(Loucaides et al., 2012). While experimental evidence on this topic is still scarce and
partly controversial  (Hervé et al., 2012; Mejía et al., 2013; Milligan et al., 2004), it is not
unlikely that OA-induced changes in the formation and dissolution of biogenic silica may
alter the strength of the frustule and therefore the palatability of diatoms to zooplankton
grazers (Friedrichs et al., 2013; Hamm et al., 2003; Liu et al., 2016; Wilken et al., 2011).
As for the other physiological effects e.g. on carbon fixation, it is likely that OA impacts
on silicification will vary among different diatoms species e.g. according to their species-
specific intrinsic buffering capacity, thereby leading to further taxonomic shifts within
diatom communities.
The response of diatoms to increasing $pCO_2$ in natural environments will be further
modified by multiple other environmental drivers changing simultaneously. Climate
change is expected to elevate ocean temperature, as well as also irradiance and nutrient
availability via changes in stratification. Physiological experiments have shown that
elevated $pCO_2$ may have beneficial effects under low and moderate irradiance, but this
effect may reverse under high light conditions due to enhanced photoinhibition (Gao
2012). Analogously, warming may have positive or negative effects on photosynthesis
and metabolism in general, depending on the thermal optima of the respective species
(Boyd et al., 2018). Altogether, these multiple additional drivers will also affect diatom
communities, leading to shifts in their taxonomic composition and size structure, which
will interact with the impacts of OA.
**4.3 Indirect $CO_2$ effects on diatom assemblages through food web interactions**


Diatom community responses can not only originate from a direct $CO_2$ effect on their
physiology but also be caused indirectly through $CO_2$ responses on other components of
the food web. For example, if a grazer of a diatom species is negatively affected by OA
then this may benefit the prey and indirectly promote its abundance. Direct OA impacts
on zooplankton communities are usually assumed to play a minor role, although there is
some experimental evidence that lower pH may have physiological effects at least on
some sensitive species or developmental stages (Cripps et al., 2016; Thor and Dupont,
2015; Thor and Oliva, 2015). Nevertheless, much of the currently available empirical
evidence indicates that zooplankton communities are affected by OA rather via bottom-
up effects, e.g. via changes in primary production or taxonomic composition of the
phytoplankton community (Meunier et al., 2017). However bottom-up effects on
zooplankton biomass, size structure, or species composition may in turn trigger feedbacks
on diatom communities, thereby leading to a feedback loop that may reinforce until a new
steady state is reached. Such considerations illustrate that also second or third order
effects need to be considered when assessing OA effects on the level of ecological
communities. Accounting for such indirect effects requires a holistic approach
considering all key players in of the food web (something that is beyond the scope of this
study). Therefore, interpretations about what the observed responses could mean for
entire plankton food webs or even biogeochemical element cycles (section 4.4) should
always be regarded with some healthy skepticism as they often neglect the potential for
indirect effects.
**4.4 Implications of changes in diatom community structure for pelagic food webs**
**and biogeochemical cycles**
The taxonomic composition and size structure of phytoplankton communities influences
the transfer of energy from primary production to higher trophic levels. In theory, larger



diatoms should support a more direct transfer because less trophic intermediates are
needed and therefore less respiration occurs until prey items are in an appropriate size
range for top predators (Azam et al., 1983; Pomeroy, 1974; Sommer et al., 2002).
Likewise a reduced abundance of the potentially toxic genus *Pseudo-Nitzschia* under high
$CO_2$ could further improve trophic transfer and growth of consumers in food webs where
*Pseudo-Nitzschia* exerts harmful impacts at present. Such changes at the bottom of a food
web might eventually lead to higher production in higher trophic levels such as fish.
Indeed, recent experimental evidence indicated that fish (including commercially
important species) could under certain constellations benefit from high $CO_2$ due to higher
food availability, although it was not tested if this response is somehow linked to the
diatom community (Goldenberg et al., 2018; Sswat et al., 2018).
Fluxes of elements through the oceans are (like fluxes of energy through food webs)
influenced by the composition of diatom communities (Sarmiento and Gruber, 2006).
This is particularly well recognized in the context of organic carbon export to the deep
ocean, for which diatoms are considered to play a pivotal role (Smetacek, 1985). Given
that high $CO_2$ favours large and perhaps more silicified diatoms over smaller ones
(section 4.2), we might expect accelerated sinking and thus a positive feedback on the
vertical carbon flux. This classical hypothesis is supported by observational evidence
from two consecutive years of the North Atlantic spring bloom where, despite similar
primary production, export was much higher in the year when the larger diatom species
dominated (Boyd and Newton, 1995). However, whether the positive relationship
between size and carbon export holds under all circumstances is by no means clear
(Tréguer et al., 2018) . It is possible that shifts towards larger sized species coincide with
shifts in other traits that feed back negatively on carbon export. For example, when the
size shift is associated with decreasing C:Si stoichiometry it may ultimately reduce carbon



export (Assmy et al., 2013).
The abovementioned examples of trophic transfer and export fluxes illustrate the
importance of the factor "diatom community structure" in the context of marine food
production and biogeochemical fluxes. They also illustrate that our understanding of the
feedbacks induced through changes in diatom communities is highly incomplete. Hence,
with our limited understanding we must currently classify $CO_2$-induced changes in
diatom communities as "a potential risk" causing changes in key ecosystem services.
**Acknowledgements**
We thank Nauzet Hernández-Hernández, Ulf Riebesell, and Javier Arístegui for their
comments on an earlier version of the data compilation. The research was funded by the
Federal Ministry of Science and Education (Bundesministerium für Bildung und
Forschung; BMBF) in the framework of the "Biological Impacts of Ocean Acidification"
project (BIOACID III, FKZ 03F0728).

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

Availability in the Western North Pacific as Determined by Next-Generation
Sequencing, PLoS One, 11(4), e0154291, doi:10.1371/journal.pone.0154291, 2016.
Fabricius, K. E., Langdon, C., Uthicke, S., Humphrey, C., Noonan, S., De'ath, G.,
Okazaki, R., Muehllehner, N., Glas, M. S. and Lough, J. M.: Losers and winners in
coral reefs acclimatized to elevated carbon dioxide concentrations, Nat. Clim. Chang.,
1(6), 165–169, doi:10.1038/nclimate1122, 2011.



Falkowski, P. G. and Raven, J. A.: Aquatic Photosynthesis, Princeton University Press,
Princeton., 2007.
Feng, Y., Hare, C. E., Leblanc, K., Rose, J. M., Zhang, Y., DiTullio, G. R., Lee, P. A.,
Wilhelm, S. W., Rowe, J. M., Sun, J., Nemcek, N., Gueguen, C., Passow, U., Benner, I.,
Brown, C. and Hutchins, D. A.: Effects of increased $pCO_2$ and temperature on the north
atlantic spring bloom. I. The phytoplankton community and biogeochemical response,
Mar. Ecol. Prog. Ser., 388, 13–25, doi:10.3354/meps08133, 2009.
Feng, Y., Hare, C. E., Rose, J. M., Handy, S. M., DiTullio, G. R., Lee, P. A., Smith, W.
O., Peloquin, J., Tozzi, S., Sun, J., Zhang, Y., Dunbar, R. B., Long, M. C., Sohst, B.,
Lohan, M. and Hutchins, D. A.: Interactive effects of iron, irradiance and $CO_2$ on Ross
Sea phytoplankton, Deep. Res. Part I Oceanogr. Res. Pap., 57(3), 368–383,
doi:10.1016/j.dsr.2009.10.013, 2010.
Ferguson, R. L., Buckley, E. N. and Palumbo, A. V.: Response of marine
bacterioplankton to differential filtration and confinement., Appl. Environ. Microbiol.,
47(1), 49–55, doi:0099-2240/84/010049-07$02., 1984.
Field, C. B., Behrenfeld, M. J., Randerson, J. T. and Falkowski, P. G.: Primary
Production of the Biosphere: Integrating Terrestrial and Oceanic Components, Science,
281(5374), 237–240, doi:10.1126/science.281.5374.237, 1998.
Flynn, K. J., Blackford, J. C., Baird, M. E., Raven, J. A., Clark, D. R., Beardall, J.,
Brownlee, C., Fabian, H. and Wheeler, G. L.: Changes in pH at the exterior surface of
plankton with ocean acidification, Nat. Clim. Chang., 2(7), 510–513,
doi:10.1038/nclimate1489, 2012.
Fogg, G. E. and Calvario-Martinez, O.: Effects of bottle size in determinations of
primary productivity by phytoplankton, Hydrobiologia, 173(2), 89–94,
doi:10.1007/BF00015518, 1989.
Friedrichs, L., Hörnig, M., Schulze, L., Bertram, A., Jansen, S. and Hamm, C.: Size and
biomechanic properties of diatom frustules influence food uptake by copepods, Mar.
Ecol. Prog. Ser., 481(2000), 41–51, doi:10.3354/meps10227, 2013.
Gao, K. and Campbell, D. A.: Photophysiological responses of marine diatoms to
elevated $CO_2$ and decreased pH: A review, Funct. Plant Biol., 41(5), 449–459,
doi:10.1071/FP13247, 2014.
Gao, K., Xu, J., Gao, G., Li, Y., Hutchins, D. A., Huang, B., Wang, L., Zheng, Y., Jin,
P., Cai, X., Häder, D. P., Li, W., Xu, K., Liu, N. and Riebesell, U.: Rising $CO_2$ and
increased light exposure synergistically reduce marine primary productivity, Nat. Clim.
Chang., 2(7), 519–523, doi:10.1038/nclimate1507, 2012.
Gazeau, F., Sallon, A., Pitta, P., Tsiola, A., Maugendre, L., Giani, M., Celussi, M.,
Pedrotti, M. L., Marro, S. and Guieu, C.: Limited impact of ocean acidification on
phytoplankton community structure and carbon export in an oligotrophic environment:
Results from two short-term mesocosm studies in the Mediterranean Sea, Estuar. Coast.
Shelf Sci., 186, 72–88, doi:10.1016/j.ecss.2016.11.016, 2017.
Giordano, M., Beardall, J. and Raven, J. A.: $CO_2$ concentrating mechanisms in algae:



mechanisms, environmental modulation, and evolution., Annu. Rev. Plant Biol., 56, 99–
131, doi:10.1146/annurev.arplant.56.032604.144052, 2005.
Goldenberg, S. U., Nagelkerken, I., Marangon, E., Bonnet, A. and Camilo, M.:
Ecological complexity buffers the impacts of future climate on marine animals
Corresponding author Keywords, Nat. Clim. Chang., 1–19, doi:10.1038/s41558-018-
619     0086-0, 2018.

Grear, J. S., Rynearson, T. A., Montalbano, A. L., Govenar, B. and Menden-Deuer, S.:
$pCO_2$ effects on species composition and growth of an estuarine phytoplankton
community, Estuar. Coast. Shelf Sci., 190, 40–49, doi:10.1016/j.ecss.2017.03.016,
623     2017.

Guangao, L.: Different types of ecosystem experiments, in Enclosed Experimental
Marine Ecosystems: A Review and Recommendations, edited by C. M. Lalli, pp. 7–20,
Springer-Verlag, New York., 1990.
Hall-Spencer, J. M., Rodolfo-Metalpa, R., Martin, S., Ransome, E., Fine, M., Turner, S.
M., Rowley, S. J., Tedesco, D. and Buia, M.-C.: Volcanic carbon dioxide vents show
ecosystem effects of ocean acidification, Nature, 454, 96–99, doi:10.1038/nature07051,
630     2008.

Hama, T., Inoue, T., Suzuki, R., Kashiwazaki, H., Wada, S., Sasano, D., Kosugi, N. and
Ishii, M.: Response of a phytoplankton community to nutrient addition under different
$CO_2$ and pH conditions, J. Oceanogr., 72(2), 207–223, doi:10.1007/s10872-015-0322-4,
634     2016.

Hamm, C. and Smetacek, V.: Armor: Why, when, and how, in Evolution of
Phytoplankton, edited by P. G. Falkowski and A. H. Knoll, pp. 311–332, Elsevier,
Boston., 2007.
Hamm, C. E., Merkel, R., Springer, O., Jurkojc, P., Maiert, C., Prechtelt, K. and
Smetacek, V.: Architecture and material properties of diatom shells provide effective
mechanical protection, Nature, 421(6925), 841–843, doi:10.1038/nature01416, 2003.
Hammes, F., Vital, M. and Egli, T.: Critical evaluation of the volumetric "bottle effect"
on microbial batch growth, Appl. Environ. Microbiol., 76(4), 1278–1281,
doi:10.1128/AEM.01914-09, 2010.
Hare, C. E., Leblanc, K., DiTullio, G. R., Kudela, R. M., Zhang, Y., Lee, P. A.,
Riseman, S. and Hutchins, D. A.: Consequences of increased temperature and $CO_2$ for
phytoplankton community structure in the Bering Sea, Mar. Ecol. Prog. Ser., 352, 9–16,
doi:10.3354/meps07182, 2007.
Hervé, V., Derr, J., Douady, S., Quinet, M., Moisan, L. and Lopez, P. J.:
Multiparametric Analyses Reveal the pH-Dependence of Silicon Biomineralization in
Diatoms, PLoS One, 7(10), e46722, doi:10.1371/journal.pone.0046722, 2012.
Hopkins, F. E., Turner, S. M., Nightingale, P. D., Steinke, M., Bakker, D. and Liss, P.
S.: Ocean acidification and marine trace gas emissions, Proc. Natl. Acad. Sci. U. S. A.,
107, 760–765, doi:10.1073/pnas.0907163107, 2010.



Hoppe, C. J. M., Hassler, C. S., Payne, C. D., Tortell, P. D., Rost, B. R. and Trimborn,
S.: Iron limitation modulates ocean acidification effects on Southern Ocean
phytoplankton communities, PLoS One, 8(11), doi:10.1371/journal.pone.0079890,
657    2013.

Hoppe, C. J. M., Schuback, N., Semeniuk, D. M., Maldonado, M. T. and Rost, B.:
Functional Redundancy Facilitates Resilience of Subarctic Phytoplankton Assemblages
toward Ocean Acidification and High Irradiance, Front. Mar. Sci., 4,
doi:10.3389/fmars.2017.00229, 2017a.
Hoppe, C. J. M., Schuback, N., Semeniuk, D., Giesbrecht, K., Mol, J., Thomas, H.,
Maldonado, M. T., Rost, B., Varela, D. E. and Tortell, P. D.: Resistance of Arctic
phytoplankton to ocean acidification and enhanced irradiance, Polar Biol., 41(3), 399–
413, doi:10.1007/s00300-017-2186-0, 2017b.
Hussherr, R., Levasseur, M., Lizotte, M., Tremblay, J. É., Mol, J., Thomas, H.,
Gosselin, M., Starr, M., Miller, L. A., Jarniková, T., Schuback, N. and Mucci, A.:
Impact of ocean acidification on Arctic phytoplankton blooms and dimethyl sulfide
concentration under simulated ice-free and under-ice conditions, Biogeosciences, 14(9),
2407–2427, doi:10.5194/bg-14-2407-2017, 2017.
James, R. K., Hepburn, C. D., Cornwall, C. E., McGraw, C. M. and Hurd, C. L.:
Growth response of an early successional assemblage of coralline algae and benthic
diatoms to ocean acidification, Mar. Biol., 161(7), 1687–1696, doi:10.1007/s00227-
674    014-2453-3, 2014.

Johnson, V. R., Brownlee, C., Rickaby, R. E. M., Graziano, M., Milazzo, M. and Hall-
Spencer, J. M.: Responses of marine benthic microalgae to elevated $CO_2$, Mar. Biol.,
160(8), 1813–1824, doi:10.1007/s00227-011-1840-2, 2011.
Kim, J. M., Lee, K., Yang, E. J., Shin, K., Noh, J. H., Park, K. T., Hyun, B., Jeong, H.
J., Kim, J. H., Kim, K. Y., Kim, M., Kim, H. C., Jang, P. G. and Jang, M. C.: Enhanced
production of oceanic dimethylsulfide resulting from $CO_2$-induced grazing activity in a
high $CO_2$ world, Environ. Sci. Technol., 44(21), 8140–8143, doi:10.1021/es102028k,
682    2010.

Kim, J.-M., Lee, K., Shin, K., Kang, J.-H., Lee, H.-W., Kim, M., Jang, P.-G. and Jang,
M.-C.: The effect of seawater $CO_2$ concentration on growth of a natural phytoplankton
assemblage in a controlled mesocosm experiment, Limnol. Oceanogr., 51(4), 1629–
1636, doi:10.4319/lo.2006.51.4.1629, 2006.
Liu, H., Chen, M., Zhu, F. and Harrison, P. J.: Effect of Diatom Silica Content on
Copepod Grazing, Growth and Reproduction, Front. Mar. Sci., 3(June), 1–7,
doi:10.3389/fmars.2016.00089, 2016.
Loucaides, S., van Cappellen, P., Roubeix, V., Moriceau, B. and Ragueneau, O.:
Controls on the Recycling and Preservation of Biogenic Silica from Biomineralization
to Burial, Silicon, 4(1), 7–22, doi:10.1007/s12633-011-9092-9, 2012.
Mallozzi, A. J., Errera, R. M., Bargu, S. and Herrmann, A. D.: Impacts of elevated
$pCO_2$ on estuarine phytoplankton biomass and community structure in two
biogeochemically distinct systems in Louisiana, USA, J. Exp. Mar. Bio. Ecol.,



511(September 2018), 28–39, doi:10.1016/j.jembe.2018.09.008, 2019.
Mann, D. G. and Vanormelingen, P.: An Inordinate Fondness ? The Number ,
Distributions , and Origins of Diatom Species, J. Eukaryot. Microbiol., 60, 414–420,
doi:10.1111/jeu.12047, 2013.
Martin-Jézéquel, V., Hildebrand, M. and Brzezinski, M. A.: Review Silicon Metabolism
in Diatoms: Implications for Growth, J. Phycol., 36, 821–840, 2000.
Maugendre, L., Gattuso, J.-P., Louis, J., de Kluijver, A., Marro, S., Soetaert, K. and
Gazeau, F.: Effect of ocean warming and acidification on a plankton community in the
NW Mediterranean Sea, ICES J. Mar. Sci., 72(6), 1744–1755,
doi:10.1093/icesjms/fsu161, 2015.
Mejía, L. M., Isensee, K., Méndez-Vicente, A., Pisonero, J., Shimizu, N., González, C.,
Monteleone, B. and Stoll, H.: B content and Si/C ratios from cultured diatoms
(*Thalassiosira pseudonana* and *Thalassiosira weissflogii*): Relationship to seawater pH
and diatom carbon acquisition, Geochim. Cosmochim. Acta, 123, 322–337,
doi:10.1016/j.gca.2013.06.011, 2013.
Menzel, D. W. and Case, J.: Concept and Design: Controlled Ecosystem Pollution
Experiment, Bull. Mar. Sci., 27(1), 1–7, 1977.
Meunier, C. L., Algueró-Muñiz, M., Horn, H. G., Lange, J. A. F. and Boersma, M.:
Direct and indirect effects of near-future $pCO_2$ levels on zooplankton dynamics, Mar.
Freshw. Res., 68(2), 373–380, doi:10.1071/MF15296, 2017.
Milligan, A. J., Varela, D. E., Brzezinski, M. A. and Morel, F. M. M.: Dynamics of
Silicon Metabolism and Silicon Isotopic Discrimination in a Marine Diatom as a
Function of $pCO_2$, Limnol. Oceanogr., 49(2), 322–329, 2004.
Nelson, D. M., Tréguer, P., Brzezinski, M. A., Leynaert, A. and Quéguiner, B.:
Production and dissolution of biogenic silica in the ocean: Revised global estimates,
comparison with regional data and relationship to biogenic sedimentation, Global
Biogeochem. Cycles, 9(3), 359–372, doi:10.1029/95GB01070, 1995.
Nielsen, L. T., Jakobsen, H. H. and Hansen, P. J.: High resilience of two coastal
plankton communities to twenty-first century seawater acidification: Evidence from
microcosm studies, Mar. Biol. Res., 6(6), 542–555, doi:10.1080/17451000903476941,
726    2010.

Nielsen, L. T., Hallegraeff, G. M., Wright, S. W. and Hansen, P. J.: Effects of
experimental seawater acidification on an estuarine plankton community, Aquat.
Microb. Ecol., 65(3), 271–285, doi:10.3354/ame01554, 2012.
Nogueira, P., Domingues, R. B. and Barbosa, A. B.: Are microcosm volume and sample
pre-filtration relevant to evaluate phytoplankton growth?, J. Exp. Mar. Bio. Ecol., 461,
323–330, doi:10.1016/j.jembe.2014.09.006, 2014.
Pan, Y., Zhang, Y., Peng, Y., Zhao, Q. and Sun, S.: Increases of chamber height and
base diameter have contrasting effects on grazing rate of two cladoceran species:
Implications for microcosm studies, PLoS One, 10(8), 1–14,





doi:10.1371/journal.pone.0135786, 2015.
Pančić, M. and Kiørboe, T.: Phytoplankton defence mechanisms: traits and trade-offs,
Biol. Rev., 92(2), 1269–1303, doi:10.1111/brv.12395, 2018.
Park, K. T., Lee, K., Shin, K., Yang, E. J., Hyun, B., Kim, J. M., Noh, J. H., Kim, M.,
Kong, B., Choi, D. H., Choi, S. J., Jang, P. G. and Jeong, H. J.: Direct linkage between
dimethyl sulfide production and microzooplankton grazing, resulting from prey
composition change under high partial pressure of carbon dioxide conditions, Environ.
Sci. Technol., 48(9), 4750–4756, doi:10.1021/es403351h, 2014.
Paul, A. J., Bach, L. T., Schulz, K.-G., Boxhammer, T., Czerny, J., Achterberg, E. P.,
Hellemann, D., Trense, Y., Nausch, M., Sswat, M. and Riebesell, U.: Effect of elevated
$CO_2$ on organic matter pools and fluxes in a summer Baltic Sea plankton community,
Biogeosciences, 12, 6181–6203, doi:10.5194/bg-12-6181/2015/, 2015.
Pomeroy, L. R.: The ocean food web - A changing paradigm, Bioscience, 24, 499–504,
749    1974.

Pomeroy, L. R., Sheldon, J. E. and Sheldon, W. M.: Changes in bacterial numbers and
leucine assimilation during estimations of microbial respiratory rates in seawater by the
precision winkler method, Appl. Environ. Microbiol., 60(1), 328–332, 1994.
Primeau, F. W., Holzer, M. and DeVries, T.: Southern Ocean nutrient trapping and the
efficiency of the biological pump, J. Geophys. Res. Ocean., 118(5), 2547–2564,
doi:10.1002/jgrc.20181, 2013.
Raven, J. A, Giordano, M., Beardall, J. and Maberly, S. C.: Algal and aquatic plant
carbon concentrating mechanisms in relation to environmental change., Photosynth.
Res., 109(1–3), 281–96, doi:10.1007/s11120-011-9632-6, 2011.
Reul, A., Muñoz, M., Bautista, B., Neale, P. J., Sobrino, C., Mercado, J. M., Segovia,
M., Salles, S., Kulk, G., León, P., van de Poll, W. H. D., Pérez, E., Buma, A. and
Blanco, J. M.: Effect of $CO_2$, nutrients and light on coastal plankton. III. Trophic
cascade, size structure and composition, Aquat. Biol., 22, 59–76, doi:10.3354/ab00585,
763    2014.

Robinson, C. and Williams, P. J. le B.: Respiration and its measurement in surface
marine waters, in Respiration in Aquatic Environments, edited by P. A. Del Giorgio and
P. J. le B. Williams, pp. 147–180, Oxford University Press, Oxford., 2005.
Roleda, M. Y., Cornwall, C. E., Feng, Y., McGraw, C. M., Smith, A. M. and Hurd, C.
L.: Effect of ocean acidification and pH fluctuations on the growth and development of
coralline algal recruits, and an associated benthic algal assemblage, PLoS One, 10(10),
1–19, doi:10.1371/journal.pone.0140394, 2015.
Rossoll, D., Sommer, U. and Winder, M.: Community interactions dampen acidification
effects in a coastal plankton system, Mar. Ecol. Prog. Ser., 486, 37–46,
doi:10.3354/meps10352, 2013.
Rost, B., Riebesell, U., Burkhardt, S. and Sültemeyer, D.: Carbon acquisition of bloom-
forming marine phytoplankton, Limnol. Oceanogr., 48(1), 55–67, 2003.



Sala, M. M., Aparicio, F. L., Balagué, V., Boras, J. A., Borrull, E., Cardelús, C., Cros,
L., Gomes, A., López-Sanz, A., Malits, A., Martínez, R. A., Mestre, M., Movilla, J.,
Sarmento, H., Vázquez-Domínguez, E., Vaqué, D., Pinhassi, J., Calbet, A., Calvo, E.,
Gasol, J. M., Pelejero, C. and Marrasé, C.: Contrasting effects of ocean acidification on
the microbial food web under different trophic conditions, Ices J. Mar. Sci., 73(3), 670–
679, doi:10.1093/icesjms/fsv130, 2015.
Sarmiento, J. L. and Gruber, N.: Ocean biogeochemical dynamics, Princeton University
Press, Princeton., 2006.
Sarmiento, J. L., Gruber, N., Brzezinski, M. A. and Dunne, J. P.: High-latitude controls
of thermocline nutrients and low latitude biological productivity, Nature, 427(6969),
56–60, doi:10.1038/nature10605, 2004.
Sarnelle, O.: Daphnia effects on microzooplankton: comparisons of enclosure and
whole-lake responses, Ecology, 78(3), 913–928, doi:10.1016/S0010-4655(02)00300-4,
789   1997.

Sarthou, G., Timmermans, K. R., Blain, S. and Tréguer, P.: Growth physiology and fate
of diatoms in the ocean: A review, J. Sea Res., 53, 25–42,
doi:10.1016/j.seares.2004.01.007, 2005.
Schulz, K. G., Riebesell, U., Bellerby, R. G. J., Biswas, H., Meyerhöfer, M., Müller, M.
N., Egge, J. K., Nejstgaard, J. C., Neill, C., Wohlers, J. and Zöllner, E.: Build-up and
decline of organic matter during PeECE III, Biogeosciences, 5, 707–718,
doi:10.5194/bg-5-707-2008, 2008.
Schulz, K. G., Bellerby, R. G. J., Brussaard, C. P. D., Büdenbender, J., Czerny, J.,
Engel, A., Fischer, M., Koch-Klavsen, S., Krug, S. A., Lischka, S., Ludwig, A.,
Meyerhöfer, M., Nondal, G., Silyakova, A., Stuhr, A. and Riebesell, U.: Temporal
biomass dynamics of an Arctic plankton bloom in response to increasing levels of
atmospheric carbon dioxide, Biogeosciences, 10, 161–180, doi:10.5194/bg-10-161-
802   2013, 2013.

Schulz, K. G., Bach, L. T., Bellerby, R., Bermudez, R., Boxhammer, T., Czerny, J.,
Engel, A., Ludwig, A., Larsen, A., Paul, A., Sswat, M. and Riebesell, U.: Phytoplankton
blooms at increasing levels of atmospheric carbon dioxide: experimental evidence for
negative effects on prymnesiophytes and positive on small picoeukaryotes, Front. Mar.
Sci., 4(64), 1–18, doi:10.3389/fmars.2017.00064, 2017.
Segovia, M., Lorenzo, M., Maldonado, M., Larsen, A., Berger, S., Tsagaraki, T.,
Lázaro, F., Iñiguez, C., García-Gómez, C., Palma, A., Mausz, M., Gordillo, F.,
Fernández, J., Ray, J. and Egge, J.: Iron availability modulates the effects of future $CO_2$
levels within the marine planktonic food web, Mar. Ecol. Prog. Ser., 565, 17–33,
doi:10.3354/meps12025, 2017.
Sett, S., Schulz, K. G., Bach, L. T. and Riebesell, U.: Shift towards larger diatoms in a
natural phytoplankton assemblage under combined high-$CO_2$ and warming conditions,
J. Plankton Res., 40(4), 391–406, doi:10.1093/plankt/fby018, 2018.
Shaik, A. U. R., Biswas, H. and Pal, S.: Increased $CO_2$ availability promotes growth of
a tropical coastal diatom assemblage (Goa coast, Arabian Sea, India), Diatom Res.,



32(3), 325–339, doi:10.1080/0269249X.2017.1379443, 2017.
Shen, C. and Hopkinson, B. M.: Size scaling of extracellular carbonic anhydrase
activity in centric marine diatoms, J. Phycol., 51(2), 255–263, doi:10.1111/jpy.12269,
821     2015.

Smetacek, V., Klaas, C., Strass, V. H., Assmy, P., Montresor, M., Cisewski, B., Savoye,
N., Webb, A., d'Ovidio, F., Arrieta, J. M., Bathmann, U., Bellerby, R., Berg, G. M.,
Croot, P., Gonzalez, S., Henjes, J., Herndl, G. J., Hoffmann, L. J., Leach, H., Losch, M.,
Mills, M. M., Neill, C., Peeken, I., Röttgers, R., Sachs, O., Sauter, E., Schmidt, M. M.,
Schwarz, J., Terbrüggen, A. and Wolf-Gladrow, D.: Deep carbon export from a
Southern Ocean iron-fertilized diatom bloom, Nature, 487(7407), 313–319,
doi:10.1038/nature11229, 2012.
Smetacek, V. S.: Role of sinking in diatom life-hystory: ecological, evolutionary and
geological significance, Mar. Biol., 84, 239–251, doi:10.1007/BF00392493, 1985.
Sommer, U., Stibor, H., Katechakis, A., Sommer, F. and Hansen, T.: Pelagic food web
confgurations at different levels of nutrient richness and their implications for the ratio
fish production:primary production, Hydrobiologia, 484, 11–20,
doi:10.1023/A:1021340601986, 2002.
Sommer, U., Paul, C. and Moustaka-Gouni, M.: Warming and ocean acidification
effects on phytoplankton - From species shifts to size shifts within species in a
mesocosm experiment, PLoS One, 10(5), 1–17, doi:10.1371/journal.pone.0125239,
838     2015.

Sournia, A., Chrétiennot-Dinet, M. J. and Ricard, M.: Marine phytoplankton: How
many species in the world ocean?, J. Plankton Res., 13(5), 1093–1099,
doi:10.1093/plankt/13.5.1093, 1991.
Spencer, M. and Warren, P. H.: The effects of habitat size and productivity on food web
structure in small aquatic microcosms, Oikos, 75(3), 419–430 [online] Available from:
http://www.jstor.org/stable/3545882?origin=crossref, 1996.
Sswat, M., Stiasny, M. H., Taucher, J., Algueró-Muñiz, M., Bach, L. T., Jutfelt, F.,
Riebesell, U. and Clemmesen, C.: Food web changes under ocean acidification promote
herring larvae survival, Nat. Ecol. Evol., 2(May), 1–5, doi:10.1038/s41559-018-0514-6,
848     2018.

Sugie, K. and Yoshimura, T.: Effects of $pCO_2$ and iron on the elemental composition
and cell geometry of the marine diatom *Pseudo-nitzschia pseudodelicatissima*
(Bacillariophyceae), J. Phycol., 49(3), 475–488, doi:10.1111/jpy.12054, 2013.
Sun, J., Hutchins, D. A., Feng, Y., Seubert, E. L., Caron, D. A. and Fu, F. X.: Effects of
changing $pCO_2$ and phosphate availability on domoic acid production and physiology of
the marine harmful bloom diatom *Pseudo-nitzschia multiseries*, Limnol. Oceanogr.,
56(3), 829–840, doi:10.4319/lo.2011.56.3.0829, 2011.
Tatters, A. O., Fu, F. X. and Hutchins, D. A.: High $CO_2$ and silicate limitation
synergistically increase the toxicity of *Pseudo-nitzschia fraudulenta*, PLoS One, 7(2),
doi:10.1371/journal.pone.0032116, 2012.



Tatters, A. O., Roleda, M. Y., Schnetzer, A., Fu, F., Hurd, C. L., Boyd, P. W., Caron, D.
A., Lie, A. A. Y., Hoffmann, L. J. and Hutchins, D. A.: Short- and long-term
conditioning of a temperate marine diatom community to acidification and warming,
Philos. Trans. R. Soc. B Biol. Sci., 368, 20120437, doi:10.1098/rstb.2012.0437, 2013.
Tatters, A. O., Schnetzer, A., Xu, K., Walworth, N. G., Fu, F., Spackeen, J. L., Sipler,
R. E., Bertrand, E. M., McQuaid, J. B., Allen, A. E., Bronk, D. A., Gao, K., Sun, J.,
Caron, D. A. and Hutchins, D. A.: Interactive effects of temperature, $CO_2$ and nitrogen
source on a coastal California diatom assemblage, J. Plankton Res., 40(2), 151–164,
doi:10.1093/plankt/fbx074, 2018.
Taucher, J., Jones, J., James, A., Brzezinski, M. A., Carlson, C. A., Riebesell, U. and
Passow, U.: Combined effects of $CO_2$ and temperature on carbon uptake and
partitioning by the marine diatoms *Thalassiosira weissflogii* and *Dactyliosolen*
*fragilissimus*, Limnol. Oceanogr., 60(3), 901–919, doi:10.1002/lno.10063, 2015.
Taucher, J., Arístegui, J., Bach, L. T., Guan, W., Montero, M. F., Nauendorf, A.,
Achterberg, E. P. and Riebesell, U.: Response of Subtropical Phytoplankton
Communities to Ocean Acidification Under Oligotrophic Conditions and During
Nutrient Fertilization, Front. Mar. Sci., 5(September), 1–14,
doi:10.3389/fmars.2018.00330, 2018.
Thoisen, C., Riisgaard, K., Lundholm, N., Nielsen, T. G. and Hansen, P. J.: Effect of
acidification on an Arctic phytoplankton community from Disko Bay, West Greenland,
Mar. Ecol. Prog. Ser., 520, 21–34, doi:10.3354/meps11123, 2015.
Thor, P. and Dupont, S.: Transgenerational effects alleviate severe fecundity loss during
ocean acidification in a ubiquitous planktonic copepod, Glob. Chang. Biol., 21(6),
2261–2271, doi:10.1111/gcb.12815, 2015.
Thor, P. and Oliva, E. O.: Ocean acidification elicits different energetic responses in an
Arctic and a boreal population of the copepod *Pseudocalanus acuspes*, Mar. Biol.,
162(4), 799–807, doi:10.1007/s00227-015-2625-9, 2015.
Tortell, P. D., DiTullio, G. R., Sigman, D. M. and Morel, F. M. M.: $CO_2$ effects on
taxonomic composition and nutrient utilization in an Equatorial Pacific phytoplankton
assemblage, Mar. Ecol. Prog. Ser., 236, 37–43, doi:10.3354/meps236037, 2002.
Tortell, P. D., Payne, C. D., Li, Y., Trimborn, S., Rost, B., Smith, W. O., Riesselman,
C., Dunbar, R. B., Sedwick, P. and DiTullio, G. R.: $CO_2$ sensitivity of Southern Ocean
phytoplankton, Geophys. Res. Lett., 35, L04605, doi:10.1029/2007GL032583, 2008.
Tréguer, P., Bowler, C., Moriceau, B., Dutkiewicz, S., Gehlen, M., Aumont, O., Bittner,
L., Dugdale, R., Finkel, Z., Iudicone, D., Jahn, O., Guidi, L., Lasbleiz, M., Leblanc, K.,
Levy, M. and Pondaven, P.: Influence of diatom diversity on the ocean biological
carbon pump, Nat. Geosci., 11(1), 27–37, doi:10.1038/s41561-017-0028-x, 2018.
Tréguer, P. J. and De La Rocha, C. L.: The World Ocean Silica Cycle, Ann. Rev. Mar.
Sci., 5(1), 477–501, doi:10.1146/annurev-marine-121211-172346, 2013.
Trimborn, S., Lundholm, N., Thoms, S., Richter, K.-U., Krock, B., Hansen, P. J. and
Rost, B.: Inorganic carbon acquisition in potentially toxic and non-toxic diatoms: the



effect of pH-induced changes in seawater carbonate chemistry., Physiol. Plant., 133(1),
92–105, doi:10.1111/j.1399-3054.2007.01038.x, 2008.
Trimborn, S., Brenneis, T., Sweet, E., Rost, B. and 10.1594/Pangaea.824406:
Sensitivity of Antarctic phytoplankton species to ocean acidification: growth, carbon
acquisition, and species interaction, Limnol. Oceanogr. , 58(3), 997–1007,
doi:10.4319/lo.2013.58.3.0997, 2013.
Trimborn, S., Brenneis, T., Hoppe, C. J. M., Laglera, L. M., Norman, L., Santos-
Echeandía, J., Völkner, C., Wolf-Gladrow, D. and Hassler, C. S.: Iron sources alter the
response of Southern Ocean phytoplankton to ocean acidification, Mar. Ecol. Prog. Ser.,
578, 35–50, doi:10.3354/meps12250, 2017.
Vrieling, E. G., Gieskes, W. W. C. and Beelen, T. P. M.: SILICON DEPOSITION IN
DIATOMS: CONTROL BY THE pH INSIDE THE SILICON DEPOSITION
VESICLE, J. Phycol., 35(3), 548–559, doi:10.1046/j.1529-8817.1999.3530548.x, 1999.
Ward, B. A., Dutkiewicz, S., Jahn, O. and Follows, M. J.: A size-structured food-web
model for the global ocean, Limnol. Oceanogr., 57(6), 1877–1891,
doi:10.4319/lo.2012.57.6.1877, 2012.
Wilken, S., Hoffmann, B., Hersch, N., Kirchgessner, N., Dieluweit, S., Rubner, W.,
Hoffmann, L. J., Merkel, R. and Peeken, I.: Diatom frustules show increased
mechanical strength and altered valve morphology under iron limitation, Limnol.
Oceanogr., 56(4), 1399–1410, doi:10.4319/lo.2011.56.4.1399, 2011.
Witt, V., Wild, C., Anthony, K. R. N., Diaz-Pulido, G. and Uthicke, S.: Effects of ocean
acidification on microbial community composition of, and oxygen fluxes through,
biofilms from the Great Barrier Reef, Environ. Microbiol., 13(11), 2976–2989,
doi:10.1111/j.1462-2920.2011.02571.x, 2011.
Wolf, K. K. E., Hoppe, C. J. M. and Rost, B.: Resilience by diversity: Large
intraspecific differences in climate change responses of an Arctic diatom, Limnol.
Oceanogr., 63(1), 397–411, doi:10.1002/lno.10639, 2018.
Wolf-Gladrow, D. and Riebesell, U.: Diffusion and reactions in the vicinity of plankton:
A refined model for inorganic carbon transport, Mar. Chem., 59(1–2), 17–34,
doi:10.1016/S0304-4203(97)00069-8, 1997.
Wu, Y., Campbell, D. A., Irwin, A. J., Suggett, D. J. and Finkel, Z. V.: Ocean
acidification enhances the growth rate of larger diatoms, Limnol. Oceanogr., 59(3),
1027–1034, doi:10.4319/lo.2014.59.3.1027, 2014.
Yoshimura, T., Nishioka, J., Suzuki, K., Hattori, H., Kiyosawa, H. and Watanabe, Y.
W.: Impacts of elevated $CO_2$ on organic carbon dynamics in nutrient depleted Okhotsk
Sea surface waters, J. Exp. Mar. Bio. Ecol., 395(1–2), 191–198,
doi:10.1016/j.jembe.2010.09.001, 2010.
Yoshimura, T., Suzuki, K., Kiyosawa, H., Ono, T., Hattori, H., Kuma, K. and Nishioka,
J.: Impacts of elevated $CO_2$ on particulate and dissolved organic matter production:
Microcosm experiments using iron-deficient plankton communities in open subarctic
waters, J. Oceanogr., 69(5), 601–618, doi:10.1007/s10872-013-0196-2, 2013.





Young, J. N., Kranz, S. A., Goldman, J. A. L., Tortell, P. D. and Morel, F. M. M.:
Antarctic phytoplankton down-regulate their carbon-concentrating mechanisms under
high $CO_2$ with no change in growth rates, Mar. Ecol. Prog. Ser., 532, 13–28,
doi:10.3354/meps11336, 2015.
Zeebe, R. E. and Wolf-Gladrow, D. A.: $CO_2$ in seawater: Equilibrium, kinetics,
isotopes, Elsevier O., Elsevier, Amsterdam., 2001.

**Tables and Figures**

**Table 1**. Response of diatom communities to high $CO_2$. 69 experiments from 54
publications were considered here. Location refers to the place where diatom
communities were collected. The RDR is dimensionless (see methods). T is the average
incubation temperature in °C. DoE are days of experiment with the number of samplings
given as the second number. Pre-filt. gives the mesh size in case the collected plankton
community was pre-filtered before incubation. Setup refers to the incubation style:
undiluted volumes (batch), repeatedly diluted volumes (s.-cont.), flow-through setups
(fl.-thr.; only benthos), chemostats (chem.; only pelagic), $CO_2$ vent sites (seep; only
benthos). Incubations can either be performed on deck (e.g. shipboards), *in situ* (e.g. *in*
*situ* mesocosms) or under laboratory conditions. V refers to the incubation volume.
Nutrient ammendments were made in some but not all studies. The element indicates
which nutrients were added. Asterisks indicate the presense of residual nutrients at the
beginning of the study. Manipulations were done with: $CO_2$ saturated seawater (SWsat),
acid additions (Acid), combined additions of acid and base (Comb.), $CO_2$ gas additions
($CO_2$), Aeration at target $p$$CO_2$ (Aer.), Passing $CO_2$ gas through a diffusive silicone tubing
(Diff.). Meth. indicates the applied methodology to investigate diatom communities: light
microscopy (LM), pigment analyses (PA), flow cytometry (FC), genetic tools (PCR),
biogenic silica (BSi). The $p$$CO_2$ range of the experiment with the number of treatments
given in brackets. The response of the bulk diatom community to $CO_2$: no effect (~),





positive (p), negative (n), not reported (N/A). The $pCO_2$ response indicates approximately
in between which treatments a $CO_2$ response was observed. Please note that this is based
on visual inspection of the datasets and therefore involves subjectivity. Please also note
that the range equals the treatment values in case only two treatments were set up. $CO_2$
induced shifts between diatom species can be: shift to larger species (large), shift to
smaller species (small), unspecified shift (shift), no species shift detected (~), not reported
(N/A). Winners or losers of the diatom community comprise: *Chaetoceros* (Chae), large
*Chaetoceros* (Chae I), medium *Chaetoceros* (Chae II), small *Chaetoceros* (Chae III),
*Neosyndra* (Neos), *Rhabdonema* (Rhab), *Eucampia* (Euca), *Cerataulina* (Cera),
*Thalassiosira* (Thals), *Proboscia* (Prob), *Pseudo-nitzschia* (Ps-n), *Thalassionema*
(Thalns), *Cylindrotheca* (Cyli), *Guinardia* (Guin), *Synedropsis* (Syned), *Dactyliosolen*
(Dact), *Toxarium* (Toxa), *Leptocylindrus* (Lept), *Grammatophora* (Gram), *Bacillaria*
(Baci), *Navicula* (Navi).

| Reference | lat | long | RDR | S | T (°C) | Habitat | DoE/ # of sampl. | Pre-filt. (µm) | Setup | Incub. | V (L) | Nutr. | Manip. | Meth. | pCO2 range (µatm) | CO2 effect | pCO2 response (µatm) | Intra-taxon effect | Winners | Losers |
|---|---|---|---|---|---|---|---|---|---|---|---|---|---|---|---|---|---|---|---|---|
| (Bach et al., 2017) | 58.264 | 11.479 | 76.2 | 29 | 7 | est. | 113/57 | 1000 | batch | in situ | 50000 | *none | SWsat | PA, LM | (2) 380, 760 | p | 380 - 760 | large | Cosc | |
| (Bach et al., 2019) | 27.990 | -15.369 | 59.6 | 37 | 18.5 | coastal | 32/21 | 3000 | batch | in situ | 8000 | N,P,Si | SWsat | LM, BSi | (7) 380 - 1120 | p | 380 - 1120 | large | Chae, Guin, Lept | Nitz |
| (Biswas et al., 2011) | 16.750 | 81.100 | 2.1 | 25 | 29.5 | est. | 5/2 | 200 | batch | Deck | 5.6 | *none/N, P | Comb. | PA | (4) 230 - 1860 | n | 650 - 1400 | N/A | | |
| (Biswas et al., 2017) | 17.000 | 83.000 | 1.5 | ? | ? | coastal | 2/1 | 200 | batch | Deck | 2 | *N,P,Si,Fe,(Zn) | Comb. | LM | (2) 230, 2200 | p | 230 - 2200 | shift | Skel | Thals |
| (Davidson et al., 2016) | -68.583 | 77.967 | 10.5 | 34 | 0.1 | coastal | 8/5 | 200 | batch | Lab | 650 | *Fe | SWsat | LM | (6) 80 - 2420 | n | 1280 - 1850 | small | Frag | Chae |
| (Domingues et al., 2017) | 37.017 | -8.500 | 7.4 | ? | 23.5 | est. | 1/1 | no | batch | Deck | 4.5 | N,P,Si,NH4 | Comb. | LM, PA | (2) 420, 710 | ~ | | ~ | | |
| (Donahue et al., 2019) | -45.800 | 171.130 | 2.6 | 34 | 11 | oceanic | 14/5 | 200 | batch | Lab | 10 | *Fe | Diff. | LM, FC | (2) 350, 620 | ~ | | N/A | | |
| (Donahue et al., 2019) | -45.830 | 171.540 | 2.6 | 34 | 11 | oceanic | 21/4 | 200 | batch | Lab | 10 | *Fe | Diff. | LM, FC | (2) 350, 630 | p | | N/A | | |
| (Eggers et al., 2014) | 38.633 | -27.067 | 1.9 | 36 | 15 | coastal | 9-10/3 | 200 | batch | Deck | 4 | N,P,Si | Comb. | LM | (2) 380, 910 | p | 380 - 910 | large | Chae III | Thals |
| (Eggers et al., 2014) | 38.650 | -27.250 | 1.9 | 36 | 15 | coastal | 9-10/4 | 200 | batch | Deck | 4 | N,P,Si | Comb. | LM | (2) 380, 910 | p | 380 - 910 | large | Thals, Chae II | Chae I |
| (Eggers et al., 2014) | 38.617 | -27.250 | 1.9 | 36 | 15 | oceanic | 9-10/5 | 200 | batch | Deck | 4 | N,P,Si | Comb. | LM | (2) 380, 910 | ~ | | N/A | | |





| | | | | | | | | | | | | | | | | | | | | |
|---|---|---|---|---|---|---|---|---|---|---|---|---|---|---|---|---|---|---|---|---|
| (Endo et al., 2013) | 46.000 | 160.000 | 2.8 | 33 | 14 | oceanic | 14/3 | 197 | batch | Deck | 12 | *none | Aer. | PA | (4) 230 - 1120 | ~ | | N/A | | |
| (Endo et al., 2015) | 53.083 | -177.000 | 2.8 | ? | 8.2 | oceanic | 5/3 | 197 | batch | Deck | 12 | *none | Aer. | PA, PCR | (2) 360, 600 | n | 360 - 600 | ~ | | |
| (Endo et al., 2016) | 41.500 | 144.000 | 2.8 | ? | 5.4 | oceanic | 3/3 | 197 | batch | Deck | 12 | *Fe | Aer. | PA, PCR | (4) 180 - 1000 | n | 350 - 1000 | shift | | |
| (Feng et al., 2009) | 57.580 | 15.320 | 1.7 | 35 | 12 | oceanic | 14/1-2 | 200 | s.-cont. | Deck | 2.7 | N,P | Aer. | LM, PA | (2) 390, 690 | p | 390 - 690 | large | Ps-n | Cyli |
| (Feng et al., 2010) | -74.230 | 179.230 | 1.7 | 34 | 0 | oceanic | 18/1-14 | 200 | s.-cont. | Deck | 2.7 | none | Aer. | LM, PA | (2) 380, 750 | ~ | | large | Chae | Cyli |
| (Gazeau et al., 2017) | 43.697 | 7.312 | 125.8 | 38 | 14 | coastal | 18/14 | 5000 | batch | in situ | 45000 | none | SWsat | PA | (6) 350 - 1250 | p | 600 - 1000 | N/A | | |
| (Gazeau et al., 2017) | 42.580 | 8.726 | 125.8 | 38 | 23 | coastal | 27/18 | 5000 | batch | in situ | 45000 | none | SWsat | PA | (6) 420 - 1250 | ~ | | N/A | | |
| (Grear et al., 2017) | 41.575 | -71.405 | 9.3 | ? | 9 | est. | 6/7 | no | chem. | Deck | 9.1 | ?none | Comb. | LM | (3) 220 - 720 | ~ | | ~ | | |
| (Hama et al., 2016) | 34.665 | 138.940 | 7.1 | ? | ? | coastal | 29/11 | 100 | batch | Deck | 400 | N,P,Si | Aer. | PA | (3) 400 - 1200 | ~ | | N/A | | |
| (Hare et al., 2007) | 56.515 | -164.730 | 6.0 | ? | 10.4 | coastal | 9-10/5 | no | s.-cont. | Deck | 2.5 | Fe,N,P, Si | Aer. | LM, PA | (2) 370, 750 | n | 370 - 750 | shift | | Cyli |
| (Hare et al., 2007) | 55.022 | -179.030 | 6.0 | ? | 10.4 | oceanic | 9-10/3 | no | s.-cont. | Deck | 2.5 | Fe | Aer. | LM, PA | (2) 370, 750 | n | 370 - 750 | N/A | | |
| (Hopkins et al., 2010) | 60.300 | 5.200 | 99.1 | ? | 10 | coastal | 21/9 | no | batch | in situ | 11000 | N, P | Aer. | LM | (2) 300, 600 | n | 300 - 600 | N/A | | |
| (Hoppe et al., 2013) | -66.833 | 0.000 | 1.9 | 34 | 3 | oceanic | 27-30/1 | 200 | s.-cont. | Lab | 4 | *none | Aer. | LM | (3) 200 - 810 | N/A | 400 - 810 | shift | Syned | Ps-n |
| (Hoppe et al., 2017b) | 71.406 | -68.601 | 1.9 | 33 | 9.5 | oceanic | 8/3 | 100 | s.-cont. | Deck | 8 | N,P,Si | Aer. | PA, LM | (2) 320, 990 | ~ | | ~ | | |
| (Hoppe et al., 2017a) | 63.964 | -60.125 | 1.9 | 32 | 7.9 | oceanic | 13-14/3 | 100 | s.-cont. | Deck | 8 | N,P,Si | Aer. | LM | (2) 300, 960 | n | 300 - 960 | shift | Frag | Ps-n |
| (Hussherr et al., 2017) | 71.406 | -70.188 | 2.6 | 33 | 4.3 | oceanic | 9/3-9 | 200 | batch | Deck | 10 | *none | Comb. | LM, PA | (6) 510 - 3300 | n | 1040 - 1620 | ~ | | |
| (James et al., 2014) | -45.639 | 170.671 | | ? | 11.6 | benthic | 42/2 | | fl.-thr. | Lab | 0 | none | Comb. | pic | (2) 400, 1250 | ~ | | N/A | | |
| (Johnson et al., 2011) | 38.417 | 14.950 | | 38 | 23.5 | benthic | 21/1 | NA | seep | in situ | 0 | none | NA | PA, LM | (3) 420 - 1600 | p | 420 - 590 | large | Toxa, Gram, Baci, Navi, Cocc | Cycl, Neos, Rhab, Nitz |
| (Kim et al., 2006) | 34.600 | 128.500 | 4.3 | ? | 14 | coastal | 14/? | 60 | batch | in situ | 150 | N,P | Aer. | LM | (3) 250 - 750 | N/A | 400 - 750 | shift | Skel | Nitz |
| (Kim et al., 2010) | 34.600 | 128.500 | 52.1 | ? | 12 | coastal | 20/22 | no | batch | in situ | 1600 | N,P,Si | SWsat/ Aer. | LM | (2) 400, 900 | ~ | | shift | Skel | Euca |
| (Mallozzi et al., 2019) | 29.241 | -90.935 | 2.4 | 12 | 21 | est. | 112/9 | 80 | s.-cont. | Lab | 20 | *none | Aer. | PA, LM | (2) 400, 1000 | ~ | | shift | Cyli | |
| (Mallozzi et al., 2019) | 29.272 | -89.963 | 2.4 | 17 | 21 | est. | 112/9 | 80 | s.-cont. | Lab | 20 | *none | Aer. | PA, LM | (2) 400, 1000 | ~ | | shift | Cyli | |
| (Maugendre et al., 2015) | 43.667 | -7.300 | 1.9 | ? | 15 | oceanic | 12/4 | 200 | batch | Deck | 4 | none | SWsat | PA | (2) 360, 630 | ~ | | N/A | | |
| (Nielsen et al., 2010) | 56.057 | 12.648 | 1.6 | 19 | 10.7 | est. | 14/4 | 175 | s.-cont. | Lab | 2.5 | *none | Acid | LM, PA | (3) 500 - 1500 | ~ | | ~ | | |
| (Nielsen et al., 2012) | -42.887 | 147.339 | 1.8 | 31 | 16 | coastal | 14/4 | 250 | s.-cont. | Lab | 2.5 | *none | Acid | LM, PA | (3) 300 - 1200 | ~ | | ~ | | |
| (Park et al., 2014) | 34.600 | 128.500 | 59.6 | ? | 17 | coastal | 19/17 | no | batch | in situ | 2400 | N,P,Si | SWsat/ Aer. | LM, PA | (6) 160 - 830 | p | 160 - 830 | N/A | Cera | |




| | | | | | | | | | | | | | | | | | | | | |
|---|---|---|---|---|---|---|---|---|---|---|---|---|---|---|---|---|---|---|---|---|
| (Paul et al., 2015) | 59.858 | 23.258 | 112.7 | 6 | 11 | est. | 46/22 | 3000 | batch | in situ | 54000 | none | SWsat | PA | (6) 370 - 1230 | p | 820 - 1000 | N/A | | |
| (Reul et al., 2014) | 36.540 | -4.600 | 3.3 | ? | 21 | coastal | 7/6 | 200 | batch | Deck | 20 | control/N,P | Aer. | LM, PA | (2) 500, 1000 | p | 500 - 1000 | large | | |
| (Roleda et al., 2015) | -45.639 | 170.671 | | 34 | 10.8 | benthic | 112/? | NA | fl.-thr. | Lab | 0.65 | none | Comb. | PA | (2) 430, 1170 | ~ | | N/A | | |
| (Rossoll et al., 2013) | 54.329 | 10.149 | 29.8 | 18 | 18 | est. | 28/7 | no | batch | Lab | 300 | N,P,Si | Aer. | LM | (5) 390 - 4000 | ~ | | N/A | | |
| (Sala et al., 2015) | 41.667 | 2.800 | 26.1 | 38 | 14 | coastal | 9/2 | no | batch | Lab | 200 | none | CO2 | LM | (2) 400, 800 | ~ | | N/A | | |
| (Sala et al., 2015) | 41.667 | 2.800 | 26.1 | 38 | 22 | coastal | 9/2 | no | batch | Lab | 200 | none | CO2 | LM | (2) 400, 800 | ~ | | N/A | | |
| (Schulz et al., 2008) | 60.267 | 5.217 | 133.7 | 31 | 10.5 | coastal | 25/18-23 | no | batch | in situ | 27000 | N,P | Aer. | PA | (3) 350 - 1050 | ~ | | N/A | | |
| (Schulz et al., 2013) | 78.937 | 11.893 | 158.5 | 34 | 3 | coastal | 30/26 - 30 | 3000 | batch | in situ | 45000 | N,P,Si | SWsat | LM, PA | (8) 185 - 1420 | ~ | | N/A | | |
| (Schulz et al., 2017) | 60.265 | 5.205 | 125.8 | 32 | 9 | coastal | 38/35 | 3000 | batch | in situ | 75000 | *N, P | SWsat | LM, PA | (8) 310 - 3050 | n | 1165 - 1425 | N/A | | |
| (Segovia et al., 2017) | 60.390 | 5.320 | 99.1 | ? | 11 | coastal | 22/9 | no | batch | in situ | 11000 | control | SWsat /Aer. | FC | (2) 300, 800 | ~ | | N/A | | |
| (Sett et al., 2018) | 54.329 | 10.149 | 49.8 | 20 | 5 | est. | 44/26 | sand filter | batch | Lab | 1400 | *none | SWsat | LM, FC | (2) 540, 1020 | ~ | | ~ | | |
| (Shaik et al., 2017) | 15.453 | 43.801 | 5.6 | 35 | 29 | coastal | 2/1 | no | batch | Deck | 2 | N,P,Si,Fe | CO2 | LM | (2) 330, 1000 | p | 330 - 1000 | ~ | | |
| (Shaik et al., 2017) | 15.453 | 43.801 | 5.6 | 36 | 29 | coastal | 9/1 | no | s.-cont. | Deck | 2 | N,P,Si,Fe | CO2 | LM | (2) 400, 1000 | p | 400 - 1000 | ~ | | |
| (Shaik et al., 2017) | 15.453 | 43.801 | 5.6 | 35 | 29 | coastal | 2/1 | no | batch | Deck | 2 | N,P,Si,Fe | CO2 | LM | (2) 240, 780 | p | 240 - 780 | ~ | | |
| (Sommer et al., 2015) | 54.329 | 10.149 | 49.8 | 20 | 9,15 | est. | 24/11 | sand filter | batch | Lab | 1400 | *none | SWsat | LM | (2) 440, 1040 | ~ | | shift | | Prob, Thaln, Guin, Ps-n, Chae |
| (Tatters et al., 2013) | -45.752 | 170.810 | 0.8 | 35 | 14 | coastal | 14/2 | 80 | s.-cont. | Lab | 0.8 | N,P,Si,Fe | Aer. | LM | (3) 230 - 570 | N/A | 400 - 570 | shift | Cosc, Ps-n | Navi, Chae |
| (Tatters et al., 2018) | 33.750 | -118.215 | 12.1 | ? | 19 | coastal | 10/1 | no | chem. | Deck | 20 | N/urea,P,Si | Aer. | LM | 380, 800 | N/A | | shift | | |
| (Taucher et al., 2018) | 27.928 | -15.365 | 97.6 | 37 | 24-22 | coastal | 60/35 | 3000 | batch | in situ | 35000 | N,P,Si | SWsat | LM, PA | (8) 350 - 1030 | p | 890 - 1030 | large | Guin | Lept |
| (Thoisen et al., 2015) | 69.217 | 53.367 | 1.4 | 33 | 3 | coastal | 8-17/6-9 | 250 | s.-cont. | Lab | 1.2 | *none | SWsat | LM | (4) 440 - 3500 | n | 440 - 900 | shift | Navi I | Navi II |
| (Tortell et al., 2002) | -6.600 | 81.017 | 7.1 | ? | ? | oceanic | 11/4 | no | s.-cont. | Deck | 4 | *none | Aer. | PA, LM | (2) 150, 750 | p | 150 - 440 | ~ | | |
| (Tortell et al., 2008) | NA | NA | 7.1 | ? | 0 | N/A | 10-18/? | no | s.-cont. | Lab | 4 | *Fe | Aer. | LM, PA | (3) 100 - 800 | p | 100 - 400 | large | Chae | Ps-n |
| (Tortell et al., 2008) | NA | NA | 7.1 | ? | 0 | N/A | 10-18/? | no | s.-cont. | Deck | 4 | *Fe | Aer. | LM, PA | (3) 100 - 800 | p | 100 - 400 | large | Chae | Ps-n |
| (Tortell et al., 2008) | NA | NA | 7.1 | ? | 0 | N/A | 10-18/? | no | s.-cont. | Deck | 4 | *Fe | Aer. | LM, PA | (3) 100 - 800 | p | 100 - 400 | large | Chae | Ps-n |
| (Trimborn et al., 2017) | -53.013 | 10.025 | 1.9 | 34 | 3 | oceanic | 30/4 | 200 | s.-cont. | Lab | 4 | none | Aer. | LM | 420, 910 | n | 420 - 910 | shift | | Ps-n |
| (Witt et al., 2011) | -23.450 | 151.917 | | ? | 24-25 | benthic | 11/4 | NA | fl.-thr. | Deck | 10 | none | SWsat | LM | (4) 310 - 1140 | p | 560 - 1140 | N/A | | |
| (Wolf et al., 2018) | 78.917 | 11.933 | 1.9 | ? | 3 | coastal | 10 - 13/1 | 200 | s.-cont. | Lab | 4 | none | Aer. | LM | (2) 400, 1000 | N/A | | ~ | | |
| (Yoshimura et al., 2010) | 49.500 | 148.250 | 2.7 | 33 | 13.5 | oceanic | 14/5 | 243 | batch | Deck | 9 | | Aer. | PA | (4) 150 - 590 | n | 150 - 280 | N/A | | |





| | | | | | | | | | | | | | | | | | | | | |
|---|---|---|---|---|---|---|---|---|---|---|---|---|---|---|---|---|---|---|---|---|
| (Yoshimura et al., 2013) | 53.390 | 177.010 | 2.8 | ? | 8.4 | oceanic | 14/3 | 197 | batch | Deck | 12 | *none | Aer. | PA, LM | 4 (300 - 1190) | p | 960 - 1190 | N/A | | |
| (Yoshimura et al., 2013) | 49.020 | 174.020 | 2.8 | ? | 9.2 | oceanic | 14/3 | 197 | batch | Deck | 12 | *none | Aer. | PA, LM | (4) 230 - 1110 | p | 880 - 1110 | N/A | | |
| (Young et al., 2015) | -44.779 | 64.073 | 7.1 | ? | -1 | coastal | 21/21 | no | s.-cont. | Deck | 4 | *none | Aer. | PA | (3) 100 - 800 | ~ | | N/A | | |
| (Young et al., 2015) | -44.780 | 64.073 | 7.1 | ? | -0.5 | coastal | 16/16 | no | s.-cont. | Deck | 4 | *none | Aer. | PA, LM | (3) 100 - 800 | ~ | | N/A | | |
| (Young et al., 2015) | -44.780 | 64.073 | 7.1 | ? | 1.5 | coastal | 20/20 | no | s.-cont. | Deck | 4 | *none | Aer. | PA | (3) 100 - 800 | ~ | | N/A | | |



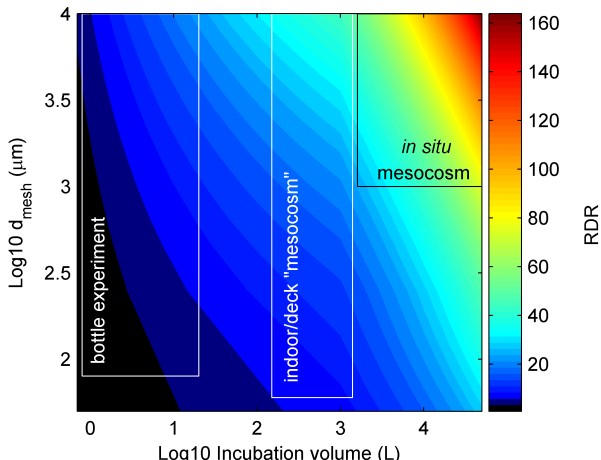


**Figure 1**. RDR as a function of incubation volume and size of the mesh that was used
while filling the incubation volumes ($d_{mesh}$). The black and white boxes illustrate
approximate ranges of the three main types of containers used in experiments. Please note
that the general definition for mesocosms are volumes >1000 L (Guangao, 1990) but
since most authors also use this term for open batch incubations with volumes between
150 – 1000 L we also stick to this term for the intermediate class.



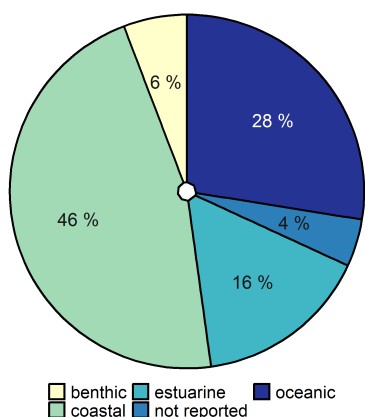


**Figure 2.** Habitats in which the ocean acidification experiments were conducted. The

total number of studies is 69. 'not reported' means that coordinates where the incubation

water was collected were not provided.

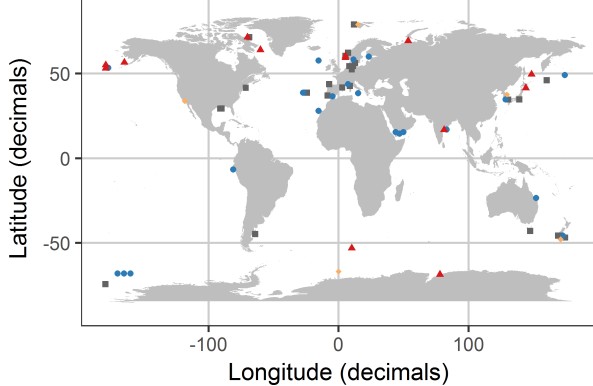


**Figure 3.** Distribution of diatom experiments with associated OA response of the bulk

communities as listed in Table 1. Blue circles = positive effect; red triangles = negative

response; grey squares = no response; orange diamonds = response not reported.

Locations were slightly modified in case of geospatial overlap to ensure visibility. Please



note that the three blue points in the Ross Sea at about -68, -165 are approximate locations
because the reference did not provide coordinates.

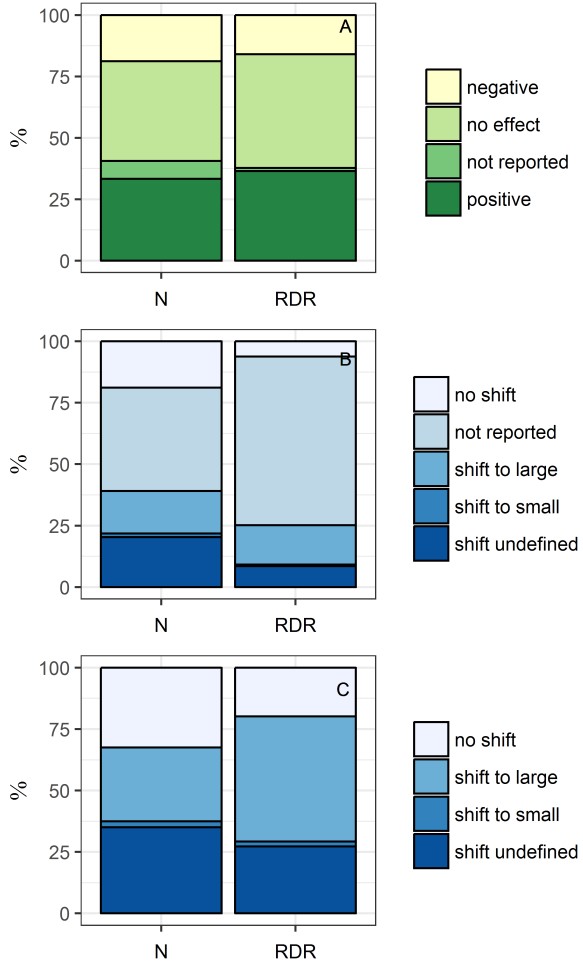


**Figure 4**. Summary of the literature analysis. (A) Response of the bulk diatom
community to ocean acidification. (B) Shifts among different diatom species due to ocean
acidification. 'Shift to large' and 'shift to small' indicate that the diatom community
shifted towards the dominance of larger or smaller species, respectively. (C) Same data
as in B but excluding studies where species shifts within the diatom community were not
reported. This reduced the dataset from 69 to 40 studies. The left column is based on the
number of studies. For example, the bulk diatom community was positively affected by
OA in 29 out of 69 studies which is 33 %. The right column is based on the RDR values.
For example, the ∑RDR value of all studies where the diatom community was positively
affected by OA was 605 which is 36 % of the total ∑RDR.

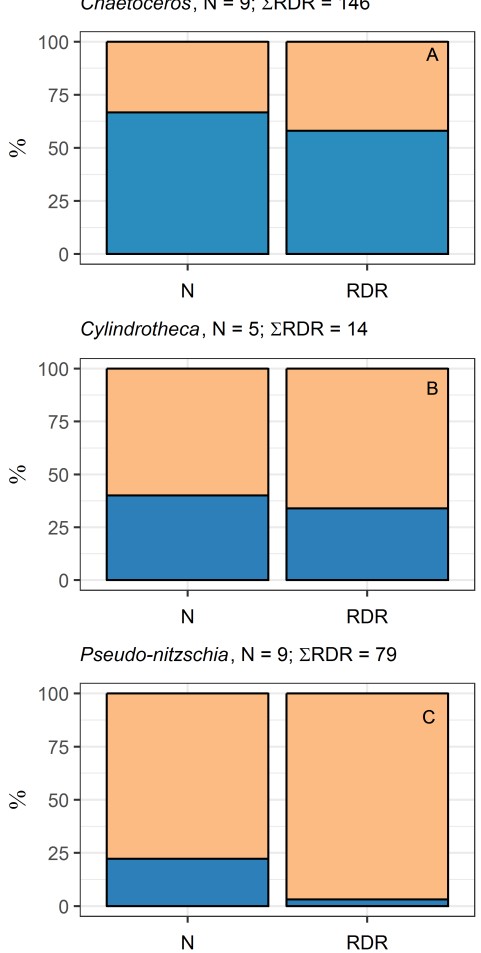


**Figure 5**. Winners and losers in diatom communities. The bar chart indicates the fraction
of experiments where the respective genera benefitted from high $CO_2$ (blue) or were put
at a disadvantage relative to the control treatment (orange). Shown here are diatom


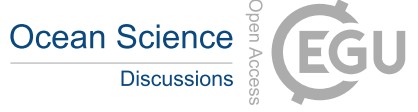

genera that were microscopically identified in at least 5 experiments. The left bars are
fractions based on the number of experiments (N, total number given above each plot).
The right bars are fractions based on the RDR values of these experiments (∑RDR of all
experiments considered given above each plot). (A) *Chaetoceros*. (B) *Cylindrotheca*. (C)
*Pseudo-nitzschia*. Please note that any such evaluation on the species level cannot be done
at present due to too few data.