# Peer review of "CO2 effects on diatoms: A Synthesis of more than a decade of ocean"

_Ocean Science, 2019_

## Referee Comment (RC1) · Anonymous Referee #1 · 27 Jun 2019

**Major points :**

Conceptually, I have major issues with the RDR score that indicates higher 'degree of realism'. If looking at the effect of CO2 on diatom physiology, I would rather trust small volume experiments where most conditions are controlled, and with the least amount of other species and grazers and which may show a direct effect on either photosynthesis, growth rates or silicification. The larger the volume, the larger the amount of uncontrolled factors that may impact diatom physiology and community shifts. Selective grazer selection of diatoms (from virus, bacteria, parasites to larger grazers) is probably the largest problem. In a large but confined mesocosm, grazer activity is in my opinion the factor that will control diatom community structure rather than CO2. The larger the volume, the larger are the probabilities of getting different organisms in the control and the various treatments that may confound interpretation. The second other confounding parameter would be competition between primary producers. In natural water experiments, the nutrient ratio for instance, together with absolute concentrations may change rapidly and induce indirect changes in community and abundances without any role of CO2. How these confounding factors are deconvoluted in large mesocosms is not clear at all (and in my view impossible). I agree that mesocosms are interesting because they take into account trophic relationships and give an insight into net effects of acidification on the total community, but I don't think their RDR scores should be considered more reliable if you're looking at direct impact of CO2 on diatoms only. What you mention on line 297 « Among these 3, only Pseudo-nitzschia was fairly consistently identified as a "loser" within the investigated natural diatom communities. The relatively weak performance of Pseudo-nitzschia spp. was somewhat surprising because previous monoclonal experiments with this genus often reported a sometimes pronounced positive (Sun et al., 2011; Tatters et al., 2012), or no influence of high CO2 on their growth rate (Sugie and Yoshimura, 2013; Trimborn et al., 2013) but more rarely a negative one (Tatters et al., 2013) » would indeed show that CO2 effects on unique strains show different results than mesocosms experiments... I would place the highest reliability of the effect of OA on diatoms on small volume experiments, which have the least amount of trophic levels, with monospecific experiments being the best, and why not competition experiment between 2 diatoms strains.

This being said, such a literature is much needed and would give valuable insights, but I don't agree at all with the data treatment method presented here, I further think this over-simplification is quite dangerous, and will likely motivate other scientists to run the same kind of "easy" methods to do literature reviews such as this one. This kind of review needs to be done carefully and with a lot of scrutiny. What we need are standardized protocols to look at OA's role on diverse organisms, and push scientists to better conceive their experimental designs, but not to make people think that we can achieve clear conclusions regarding a group as diverse as diatoms from all kinds of mixed experiments, most of which were probably not designed to investigate diatoms specifically.

I would welcome this review of papers, but the data needs to be presented more objectively. It is a good idea to regroup experiments by incubation volumes - small volume (1-4 L), middle volume (5-20 L), minicosms, mesocosms- and presence or not of grazers (through mesh size), but not with this weighing method.

**Minor points :**

line 24: what exactly do you lean by positively? higher growth rates, abundance, biomass ?

line 53 : since the title is « ocean acidification » I would expect an estimate of the number of marine diatom species here, not total. Sournia et al 91 recognized between 1,400 and 1,800 marine species, more recently the Tara-Oceans metabarcode data revealed up to 4,748 operational taxonomic units (OTU) (Malviya et al 2016), so 30,000 species seem a bit high and probably includes freshwater species ?

line 55 : you may write <2 μm, (e.g. *Minutocellus* and *Minidiscus* genera)

line 199/203 : I don't understand why you would assume spherical shape for large volume incubators, most are cylinder shaped, with a cone at the base or not depending on the model. I have not yet seen one spherical large volume incubator.

line 263 : please correct « ,. »

line 263 : I believe bibliographical references are not correctly annotated in the text, example : « microscopy except for (Endo et al., 2015) who used molecular tools »

Table 1 : I find the text too small to read, in particular if printed.

---

## Referee Comment (RC2) · Anonymous Referee #2 · 5 Jul 2019

General comment: I have reviewed this manuscript previously and I see there are many changes relative to its earlier version. I really like this idea to summarize the responses of the natural diatom assemblage from Ocean Acidification experiments and this kind of review will help to improve our current understanding and hence will be definitely helpful in modifying future experimental plans.

However, I am only afraid that the number of studies that are considered here (69) are too small with quite large variability in the protocol used for different experiments. This may lead to wrong interpretation of the results. For example, the authors identified Pseudo-Nitzschia as a looser which is highly contradictory to the existing literature on

monospecific culture. Moreover, we have conducted some onboard incubation experiments (manuscript under preparation) in the tropical waters and noticed the opposite trend. We found that Nitzschia and Pseudo-Nitzschia are dominating species under increasing CO2 levels. In the paper cited here by the authors (Biswas et al. 2017), the plots for community composition showed that Pseudo-Nitzschia abundance increased and there was no sign of decrease under high CO2 levels. Therefore, I feel that drawing a conclusion based on a limited number of studies could be largely misleading. And most importantly, the community shift in relation to increasing/decreasing CO2 levels can be largely dependent on the initial community that is used for the experiment. The paper by Eggers et al (Global change biology, 20(3), 713-723) very clearly demonstrated that the initial community is a key driving force rather than CO2 in the incubations experiments with natural community. Further, "different experimental volume" can be a major factor that finally controls the community shift. I am not sure if it would be logical to generalize the responses of the community under very different experimental exposure which would definitely neglect the bottle effect. Moreover, the total number of experiments considered here are only 69 including open ocean (28%), coastal (46%), estuarine (16%) and benthic (6%). My guestion is, can we compare the responses of the diatoms from open ocean and estuarine region, since they have quite different physiology. The former group is never exposed to high CO2 and the later is well acclimatized to a large range of pH variability. I am not sure if this would be really logical to put them in the same scale for a comparison. Pseudo-Nitzschia from open ocean region can be CO2 sensitive, whereas, the same genera from a coastal or estuarine region can be highly resilient. If we do such comparison, then the responses need to be discussed considering their background.

Considering the above points, I feel that the manuscript need major revision with a better focus and hence cannot be recommended for publication in its current form.

---

## Author Comment (AC1) · 19 Jul 2019

Dear Dr. Chapman,

we would like to thank you and the 2 anonymous reviewers for their comments which helped us to improve our manuscript. Please find a point-by-point responses to all comments in this document. The line numbers mentioned by you and the reviewers refer to the original version of the manuscript while the line numbers in our responses refer to the revised version of the manuscript.

EDITOR:

Comment 1: This is an easy-to-read paper that synthesizes a large number of experiments on the effects of increasing CO2 on diatom communities. Unfortunately, as the authors state, there is no clear standard for experimental design in such experiments, so it is hard to draw definitive conclusions. This is particularly the case as regards the size of any incubation chamber, where wall effects are known to be important and nutrient depletion may affect results, although large diatoms seem to profit more than smaller ones. It will certainly be of interest to researchers in the field.

REPLY: We thank the Editor for the kind words.

Comment 2: Readers may be somewhat confused by the determination of the "relative degree of realism" (RDR) that the authors use. Some of the writing in this section is a little clumsy, especially lines 196-199, and I could find no reference to this statement in Ferguson et al., 1984, even though it is referenced here.

REPLY: We reformulated many parts of this section to hopefully be clearer in the revised version of the manuscript (e.g. lines 176 – 191). In line 180 we cite the Ferguson et al. (1984) paper for their finding that confinement of natural bacteria communities in 4 L bottles shifted the community composition towards culturable species. This certainly is an experimental artefact due to confinement so we think this citation is correct. In line 212 we cite the Ferguson et al. (1984) paper for their finding that differential filtration of the community before confinement altered bacteria growth rates and community composition (they tested 3 $\mu$m vs 0.2 $\mu$m filtration). They write: "The 3.0 $\mu$m filtration increased growth rate and ultimate numbers of culturable cells through the removal of bacterial predators or the release of primary amines from cells damaged during filtration or both". We think that also this citation is correct. In line 222 we cited the Ferguson et al. (1984) paper for the idea that bottle effects should decrease with increasing V/S. This is only implicit in their paper so that we remove the quotation here.

Comment 3: Also, why does RDR depend on the cube root of the applied mesh size? I can understand why the authors want to reduce the RDR as mesh size decreases, but

this definition seems completely arbitrary.

REPLY: The rationale for calculating the cube root of dmesh was that in this case the influence of V/S and dmesh on RDR becomes roughly similar. We added this information to the revised version of the manuscript (lines 249 - 250).

Comment 4: Regarding the figures, Fig. 2 seems unnecessary as all the numbers are stated in the text, and Fig. 4B is not actually discussed; lines 263-264 and the rest of the paragraph actually discuss Fig 4C.

REPLY: We took out Fig. 2 from the revised version of the manuscript. Fig. 4B is indeed not discussed but is nevertheless important for completeness because it shows the fraction of studies that did not report changes in community composition. Fig. 4C is better for the discussion as it more clearly illustrates the specific changes in community composition that have been observed when this was investigated.

REVIEWER #1:

Major points : Comment 5: Conceptually, I have major issues with the RDR score that indicates higher 'degree of realism'. If looking at the effect of CO2 on diatom physiology, I would rather trust small volume experiments where most conditions are controlled, and with the least amount of other species and grazers and which may show a direct effect on either photosynthesis, growth rates or silicification. The larger the volume, the larger the amount of uncontrolled factors that may impact diatom physiology and community shifts. Selective grazer selection of diatoms (from virus, bacteria, parasites to larger grazers) is probably the largest problem. In a large but confined mesocosm, grazer activity is in my opinion the factor that will control diatom community structure rather than CO2. The larger the volume, the larger are the probabilities of getting different organisms in the control and the various treatments that may confound interpretation.

REPLY: The goal of our study was not to investigate the physiological response of diatoms to OA. Such meta analyses have already been made (Dutkiewicz et al., 2015). Our goal was to summarize how diatoms respond to OA in their natural habitat. More generally, experiments with natural ecological communities (as in our study) do not so much aim for a mechanistic understanding of a certain process (as e.g. in physiological experiments) but rather assess the general sensitivity of more natural communities to environmental drivers. Therefore, (as also noted by the Reviewer below) it is important to have a realistic setup because the net response is composed of a direct physiological response to CO2 AND by CO2-induced alterations of interactions with other species. It is therefore, in our opinion, not desirable to exclude important ecosystem components, even if they make interpretations of the results more difficult. While we agree with Reviewer #1 that grazers largely control diatom community structure, the differences in diatom community structure between treatments must still be due to CO2 because CO2 is the only factor different between treatments. Therefore, if CO2 effects on diatoms are indirectly mediated, e.g. by changes in competition with other phytoplankton or via changes in the grazer community, it is likely that these factors are also altered by OA in the real ocean. In our analysis we are interested in the differences between treatments and not primarily in the factors that mostly control diatom community structure as such. Indeed, we extensively discuss in section 4.2 how CO2 can induce the observed responses in diatom community structure and highlight the potential role of grazers.

Comment 6: The second other confounding parameter would be competition between primary producers. In natural water experiments, the nutrient ratio for instance, together with absolute concentrations may change rapidly and induce indirect changes in community and abundances without any role of CO2. How these confounding factors are deconvoluted in large mesocosms is not clear at all (and in my view impossible).

REPLY: As indicated in the previous reply, a major prerequisite of these kind of experiments is that the initial conditions in all treatments within any given experiment are as identical as possible. Like for grazers, we agree that the nutrient ratio may also

play an important role in influencing the diatom community composition as such but this is not the point here. Important for our analysis is that we compare control and high CO2 treatments and in this case CO2 must be the driving force for the observed difference because all other aspects are the same in between treatments. How exactly CO2 induces the differences between control and treatment can then differ depending on the physiological CO2 responsiveness of the diatom community and the ecological settings in the experiment. However, the same applies in nature, where CO2 increases but other environmental factors (such as nutrient ratios, but also temperature, light etc.) will strongly differ among regions. The point of our study was to account for this natural variability when assessing the direction and magnitude of CO2 effects on diatom communities – even though it is not possible to deconvolute this (as mentioned by the Reviewer).

Comment 7: I agree that mesocosms are interesting because they take into account trophic relationships and give an insight into net effects of acidification on the total community, but I don't think their RDR scores should be considered more reliable if you're looking at direct impact of CO2 on diatoms only.

REPLY: We are not looking at the direct (i.e. the physiological) impact of CO2 on diatoms but on the net response within their natural habitat. This includes direct physiological effects as well as indirect effects that are mediated e.g. via competition and trophic interactions. For this purpose, setups yielding a higher RDR score should provide a more reliable picture of the diatom response to OA in the given environment because they arguably (section 2.3) simulate the natural habitat more realistically (although we acknowledge that also larger volumes are associated with artefacts, lines 200 - 207).

Comment 8: What you mention on line 297 Âń Among these 3, only Pseudo-nitzschia was fairly consistently identified as a "loser" within the investigated natural diatom communities. The relatively weak performance of Pseudo-nitzschia spp. was somewhat surprising because previous monoclonal experiments with this genus often reported a

sometimes pronounced positive (Sun et al., 2011; Tatters et al., 2012), or no influence of high CO2 on their growth rate (Sugie and Yoshimura, 2013; Trimborn et al., 2013) but more rarely a negative one (Tatters et al., 2013) Âż would indeed show that CO2 effects on unique strains show different results than mesocosms experiments.

REPLY: The Pseudo-Nitzschia result was taken out from the manuscript (see comment 20). Nevertheless, we want to emphasize that CO2 effects can play out differently in a monoclonal culture than in a natural habitat. In fact, this is one important point that we want to emphasize with our study (lines 485 – 494). In the natural habitat, CO2 effects can be induced indirectly through CO2 effects on competitors and/or grazers. These CO2 effects on competitors and/or grazers can then lead to a chain reaction of trophic changes at the end of which the tested diatom would show a different response to what we would have expected on a purely physiological response observed in a monoclonal cell culture. We have discussed this in sections 4.1 and 4.2.

Comment 9: I would place the highest reliability of the effect of OA on diatoms on small volume experiments, which have the least amount of trophic levels, with monospecific experiments being the best, and why not competition experiment between 2 diatoms strains.

REPLY: We disagree with the Reviewer's conclusion. We think that the message must not be that smaller scale incubations with no/little ecological context are our best tool to project diatom responses to OA. Stephen Carpenter, who without doubt has the authority to make these statements, has written a seminal paper with the title: "Microcosm experiments have limited relevance for community and ecosystem ecology" (Carpenter, 1996). Based on insights from 'whole lake experiments' Carpenter nicely illustrates that: "Without context of appropriately scaled field studies, microcosm experiments become irrelevant and diversionary". Furthermore, Carpenter concludes: "Misleading inferences are greatly reduced by scaling research tools to the spatial and temporal extent of ecological processes". Accordingly, we need experiments on different scales, ranging from small-scale monoclonal experiments to large-volume mesocosm studies,

and even field studies, to get a reliable picture of how natural systems function and assess their sensitivity to environmental drivers.

Comment 10: This being said, such a literature is much needed and would give valuable insights, but I don't agree at all with the data treatment method presented here, I further think this over- simplification is quite dangerous, and will likely motivate other scientists to run the same kind of "easy" methods to do literature reviews such as this one. This kind of review needs to be done carefully and with a lot of scrutiny. What we need are standardized protocols to look at OA's role on diverse organisms, and push scientists to better conceive their experimental designs, but not to make people think that we can achieve clear conclusions regarding a group as diverse as diatoms from all kinds of mixed experiments, most of which were probably not designed to investigate diatoms specifically.

REPLY: As argued in the previous replies, we think that an over-simplification would be not to account for the different scales of experiments at all. Nevertheless, our analysis includes not only the RDR weighted approach but also the simpler type of analysis where we just counted the number of outcomes and added them to yield a cumulative score. Both analyses ultimately lead to the same conclusions so that the RDR approach does not lead to any major surprises. To clarify our stand point we added the following section to the revised version of the manuscript: "Finally, we want to point out (and explicitly acknowledge) that the RDR approach to balance the influence of studies on the final outcome of the literature analysis is of course not the one perfect solution and most likely incomplete (see above). However, balancing a literature analysis with the RDR score may still be an improvement relative to the other case where each experiment is treated exactly equal despite huge differences in the experimental setup. Nevertheless, to account for both views (i.e. the RDR is useless vs. the RDR is useful) we will present the outcome of our literature analysis in two different ways throughout the paper: 1) by simply counting the number of outcomes and adding them to yield a cumulative score (N-based approach; left columns in Figs. 3 and 4); 2) by adding

the RDR score of the experiments with a certain outcome to yield a cumulative score (RDR-based approach; right columns in Figs. 3 and 4). "

Comment 11: I would welcome this review of papers, but the data needs to be presented more objectively. It is a good idea to regroup experiments by incubation volumes - small volume (1-4 L), middle volume (5-20 L), minicosms, mesocosms- and presence or not of grazers (through mesh size), but not with this weighing method.

REPLY: We have tried to also present results in this incremental manner suggested by the Reviewer but the number of studies is often too low within the same increment to get to any useful conclusions. We also want to point out that grazers cannot be excluded through filtration because grazers are often within the same size class of diatoms (e.g. ciliates).

Minor points:

Comment 12: line 24: what exactly do you lean by positively? higher growth rates, abundance, biomass ?

REPLY: Good point. We meant positively with respect to abundance and biomass. We added this information to the revised version of the manuscript (line 29).

Comment 13: line 53 : since the title is Âń ocean acidification Âż I would expect an estimate of the number of marine diatom species here, not total. Sournia et al 91 recognized between 1,400 and 1,800 marine species, more recently the Tara-Oceans metabarcode data revealed up to 4,748 operational taxonomic units (OTU) (Malviya et al 2016), so 30,000 species seem a bit high and probably includes freshwater species ?

REPLY: We thank the Reviewer for pointing this out. The number included freshwater species. We modified this part to be clearer and included the references mentioned by the Reviewer (lines 60 - 66).

Comment 14: line 55 : you may write <2 $\mu$m, (e.g. Minutocellus and Minidiscus genera)

REPLY: We googled for the minimum sizes of Minutocellus and Minidiscus and could not find a reference clearly showing they (or some species of these genera) are smaller 2 $\mu$m. We found smaller 3 but not smaller 2 $\mu$m. We also frequently observed Minidiscus and Arcocellulus in mesocosm studies but neither of them was smaller 2 $\mu$m (as we checked with scanning electron microscopy; e.g. Bach et al., 2017). We therefore would prefer to stick to our more conservative estimate.

Comment 15: line 199/203 : I don't understand why you would assume spherical shape for large volume incubators, most are cylinder shaped, with a cone at the base or not depending on the model. I have not yet seen one spherical large volume incubator.

REPLY: The problem of unknown shape does not only apply for mesocosms but also for all the small scale incubations (bottles). Because container shape is not given in most studies (in particular not in the bottle experiments) we assumed the simplest shape (sphere) for all incubations. We justified this in the revised version of the manuscript by stating: "The assumption of spherical shape was necessary because it allowed us to calculate V/S from only knowing V which is usually the only parameter provided with respect to container characteristics. We are aware that this is a simplification because the majority of containers used in experiments will likely have had cylindrical shape. However, the conversion from volume to surface assuming cylindrical shape would have required knowledge of two dimensions (radius and height of the cylinder). Although shape can influence processes within the container (Pan et al., 2015), it is a less important factor to consider in our study because sensitivity calculations assuming reasonable cylinder dimensions showed that the V/S differences due to container shape will be small compared to the V/S differences due to the range of container volumes compared here." (Lines 227 – 236)

Comment 16: line 263 : please correct Âń ,. Âż

REPLY: We thank the Reviewer for spotting this mistake and changed accordingly.

Comment 17: line 263 : I believe bibliographical references are not correctly annotated

in the text, example : Âń microscopy except for (Endo et al., 2015) who used molecular tools Âż

REPLY: We thank the Reviewer for spotting this mistake and changed accordingly. (Also elsewhere).

Comment 18: Table 1 : I find the text too small to read, in particular if printed.

REPLY: The format of this table will be adjusted by the publisher's editing team if the paper is accepted.

REVIEWER #2:

Comment 19: General comment: I have reviewed this manuscript previously and I see there are many changes relative to its earlier version. I really like this idea to summarize the responses of the natural diatom assemblage from Ocean Acidification experiments and this kind of review will help to improve our current understanding and hence will be definitely helpful in modifying future experimental plans.

REPLY: We thank the Reviewer for the kind words.

Comment 20: However, I am only afraid that the number of studies that are considered here (69) are too small with quite large variability in the protocol used for different experiments. This may lead to wrong interpretation of the results. For example, the authors identified Pseudo-Nitzschia as a looser which is highly contradictory to the existing literature on monospecific culture. Moreover, we have conducted some onboard incubation experiments (manuscript under preparation) in the tropical waters and noticed the opposite trend. We found that Nitzschia and Pseudo-Nitzschia are dominating species under increasing CO2 levels. In the paper cited here by the authors (Biswas et al. 2017), the plots for community composition showed that Pseudo-Nitzschia abundance increased and there was no sign of decrease under high CO2 levels. Therefore, I feel that drawing a conclusion based on a limited number of studies could be largely misleading.

REPLY: We understand the Reviewer's concerns and removed the results/discussion on specific taxa from the revised version of the manuscript.

Comment 21: And most importantly, the community shift in relation to increasing/decreasing CO2 levels can be largely dependent on the initial community that is used for the experiment. The paper by Eggers et al (Global change biology, 20(3), 713-723) very clearly demonstrated that the initial community is a key driving force rather than CO2 in the incubations experiments with natural community.

REPLY: Yes, the CO2 response depends on the tested community enclosed at the beginning of each experiment. We discuss in detail how CO2 responses can be expressed (through a "direct" physiological response or "indirectly" through altered ecological interactions (see sections 4.1 and 4.2)). We would like to emphasize, however, that all controls and treatments of the investigated experiments initially had enclosed the same communities. Thus, the (significant) differences that occurred in the course of an experiment could only be caused by CO2. We are interested in this difference and not if another factor is more or less important in determining diatom community composition. (See also our replies to comments 5 and 6).

Comment 22: Further, "different experimental volume" can be a major factor that finally controls the community shift. I am not sure if it would be logical to generalize the responses of the community under very different experimental exposure which would definitely neglect the bottle effect.

REPLY: We agree with the Reviewer which is why we came up with the RDR approach to balance the influence of smaller and larger volume incubations as they are likely associated with a different degree of bottle effects. We point out in the revised manuscript that results from a simple cumulative approach AND the RDR approach are shown in this analysis (lines 279 – 289).

Comment 23: Moreover, the total number of experiments considered here are only 69 including open ocean (28%), coastal (46%), estuarine (16%) and benthic (6%). My

question is, can we compare the responses of the diatoms from open ocean and estuarine region, since they have quite different physiology. The former group is never exposed to high CO2 and the later is well acclimatized to a large range of pH variability. I am not sure if this would be really logical to put them in the same scale for a comparison. Pseudo-Nitzschia from open ocean region can be CO2 sensitive, whereas, the same genera from a coastal or estuarine region can be highly resilient. If we do such comparison, then the responses need to be discussed considering their background.

REPLY: We agree with the Reviewer that diatoms may have different sensitivities to OA depending on whether or not they originate from an environment with highly variable CO2 concentrations. We added an analysis where we separately looked at the bulk diatom response in near shore habitats (coastal + estuarine + benthic) as opposed to oceanic habitats. Our analysis indicates that oceanic diatom communities respond more frequently to high CO2 than communities from near shore habitats. This is in line with the common notion that OA could have larger effects on open ocean communities (Duarte et al., 2013). We added this analysis to the results (lines 325 – 345, Fig. 4) and discussed them in the revised version of the manuscript (lines 362 – 377).

References

Carpenter, S. R.: Microcosm Experiments Have Limited Relevance for Community and Ecosystem Ecology, Ecology, 77(3), 667–680, 1996.

Duarte, C. M., Hendriks, I. E., Moore, T. S., Olsen, Y. S., Steckbauer, A., Ramajo, L., Carstensen, J., Trotter, J. A. and McCulloch, M.: Is Ocean Acidification an Open-Ocean Syndrome? Understanding Anthropogenic Impacts on Seawater pH, Estuaries and Coasts, 36(2), 221–236, doi:10.1007/s12237-013-9594-3, 2013.

Dutkiewicz, S., Morris, J. J., Follows, M. J., Scott, J., Levitan, O., Dyhrman, S. T. and Berman-Frank, I.: Impact of ocean acidification on the structure of future phytoplankton communities, Nat. Clim. Chang., 5(11), 1002–1006, doi:10.1038/nclimate2722, 2015.

---

## Author Comment (AC2) · 19 Jul 2019

Please check replies to Reviewer #1 for our responses to all comments.

———————————————

---

## Author Response (AR2)

Dear Dr. Chapman,
thank you for you positive response. We added a paragraph with the suggested
statement to the end of section 2.1 (see lines 125 – 147 in this file). Changes made are
marked in yellow in the manuscript version with "tracked changes". We would like to
thank you very much for handling this manuscript.
Kind regards,
Lennart Bach

[revised manuscript text omitted]